# Dissecting Transformers: A 'CLEAR' Perspective towards Green AI

## Abstract

The rapid adoption of Large Language Models (LLMs) has raised significant environmental concerns. Unlike the one-time cost of training, LLM inference occurs continuously at a global scale and now dominates the AI energy footprint. Yet, most sustainability studies report only coarse, model-level metrics due to the lack of fine-grained measurement methods, treating energy efficiency more as an afterthought than as a primary objective. We present the first fine-grained empirical analysis of inference energy across core components of transformer architecture. We propose a novel methodology, **Component-Level Energy Assessment via Repetitions (CLEAR)**[1], to overcome temporal mismatch between microsecond($\mu$ s) scale component execution and monitoring of millisecond(ms) scale energy sensors. Using CLEAR, we evaluate 15 models spanning four distinct architecture types and consistently keep component-wise energy variance below **9.5%** while capturing more than **90%** of the model's total energy as individual components. Our empirical analysis reveals that Attention blocks consume significantly more energy per floating-point operation (FLOP), indicating that energy consumption is not proportionally aligned with FLOP counts. This shows that FLOPs alone fail to capture the true energy cost at a component level. Our findings establish detailed component-level energy baselines and provide insight as an initial step to build energy-efficient transformer models through component-level optimizations.

## 1 Introduction

Large language models (LLMs) such as GPT-4, Gemini, and Claude have transformed natural language processing, but their environmental costs are increasingly concerning. ChatGPT-4o alone has been estimated to produce about 150,000 tons of $CO_2$e in 2025, equivalent to annual emissions of 30,000 gasoline powered cars or the carbon sequestration of a forest the size of city of Chicago (Jegham et al., 2025). Beyond emissions, inference consumes significant energy and water, Google reports that a single Gemini query uses up to nine seconds of television viewing, with an associated reliance on water-intensive cooling that strains local supplies (News, 2025; Google, 2025). While training and finetuning of LLMs is a computationally heavy and energy intensive process, it occurs infrequently. Inference, by contrast, happens continuously at a massive scale, with GPT models serving hundreds of millions of queries daily (TechTarget, 2025; AceCloud, 2024). This makes reduction of per inference energy consumption a primary objective for model's performance optimization.

Current research focuses on model-level energy consumption, enabling high-level comparisons across systems (Alizadeh and Castor, 2024; Sánchez-Mompó et al., 2025a). However, such aggregate reporting obscures which architectural components e.g., Attention, or MLP blocks, drive energy usage. Fine-grained energy measurement is essential for identifying energy-intensive components, enabling targeted optimizations, and informing algorithm–hardware co-design for sustainable deployment.

In this paper, we introduce CLEAR, a methodology for fine-grained energy measurement of individual component blocks of the Transformer architecture during inference. Unlike

---

[1]Code: https://anonymous.4open.science/r/CLEAR-D487

prior work limited to model-level reporting, our approach decomposes transformers into constituent components, such as Embedding layer, Normalization blocks, Attention, and feed-forward MLP and measures the energy consumed by each component. This allows us to establish detailed energy baselines and to compare consumption patterns across model architectures, sizes, input lengths, and floating point precisions. A central challenge arises from the temporal granularity mismatch between component execution and energy sensor monitoring as the transformer sub-operations complete within microseconds, whereas energy sensor provides power updates at tens of milliseconds. To address this, CLEAR employs an amplification strategy that stabilizes energy measurements. This design circumvents limitations of current monitoring infrastructure and achieves component-level isolation at a granularity previously inaccessible. Our contributions can be summarized as follows

1) We propose a novel methodology, CLEAR (Component-Level Energy Assessment via Repetitions), to overcome temporal mismatch between execution of microsecond ($\mu s$) scale components and milisecond (ms) scale energy sensors. CLEAR is deviced to measure energy consumption of fine-grained components of Transformer architecture.

2) Using CLEAR, we measure component-level energy consumption for 15 models across different model architectures with high consistency and completeness and perform an empirical analysis across different input token lengths and floating point precisions.

3) Through our empirical analysis, we observe that energy consumed per FLOP of computation varies significantly across components, with Attention Block exhibiting the highest energy consumed per FLOP. Empirically, we observe that energy consumption of components can be decomposed into fixed overheads and FLOP-dependent costs enabling future research to estimate it.

## 2 Related Work

The environmental footprint of large language models (LLMs) has become a central research concern, with many studies quantifying the carbon emissions of training and inference. LLMCarbon Faiz et al. (2024) provides one of the most comprehensive analyses to date, modeling the end-to-end footprint of both dense and mixture-of-experts architectures across training, inference, experimentation, and storage. Vidur Özcan et al. (2025) complements this with GPU-based simulations, showing how batch size, sequence length, and parallelism affect inference efficiency. Fernandez et al. (2025) extend this to real workloads, distinguishing prefill and autoregressive decoding, and demonstrate that optimizations such as speculative decoding, kernel compilation, and continuous batching can cut energy use by up to 73%. Broader benchmarks, like *How Hungry is AI?* Jegham et al. (2025), evaluate energy, water, and $CO_2$ footprints across hardware platforms, while the BLOOM case study Luccioni et al. (2023) was among the first to track emissions during the training and inference of a 176B parameter model. Collectively, these efforts have established robust methodologies for characterizing the environmental impact of LLMs, but they report energy only at the aggregate model level due to the absence of methods for fine-grained measurement, leaving open the question of how energy is distributed across internal components

To improve accessibility, lightweight tools have emerged. CodeCarbon estimates emissions from GPU/CPU usage and regional carbon intensity, while Carbontracker Anthony et al. (2020) adds real-time monitoring and early prediction of training costs. These tools improve transparency but remain oriented toward system-level aggregation. Recent fine-grained approaches like Rajput et al. (2024) introduced FECoM, profiling TensorFlow APIs via static instrumentation to build one of the first framework-level energy datasets. Pinnock et al. (2025) proposed EDGEPROFILER, an analytical framework for lightweight LLMs on devices such as Raspberry Pi and Jetson Nano, showing that quantization can reduce memory use by 70% and energy by 50%, but that I/O bottlenecks often dominate latency. Hugging Face's AI Energy Score Luccioni and collaborators (2025) benchmarks over 160 models on 10 tasks, reporting GPU energy use across preprocessing, prefill, and decode stages.

Overall, existing studies neither provide a detailed empirical analysis of fine-grained, component-level energy consumption nor offer a reliable way to measure energy at such a

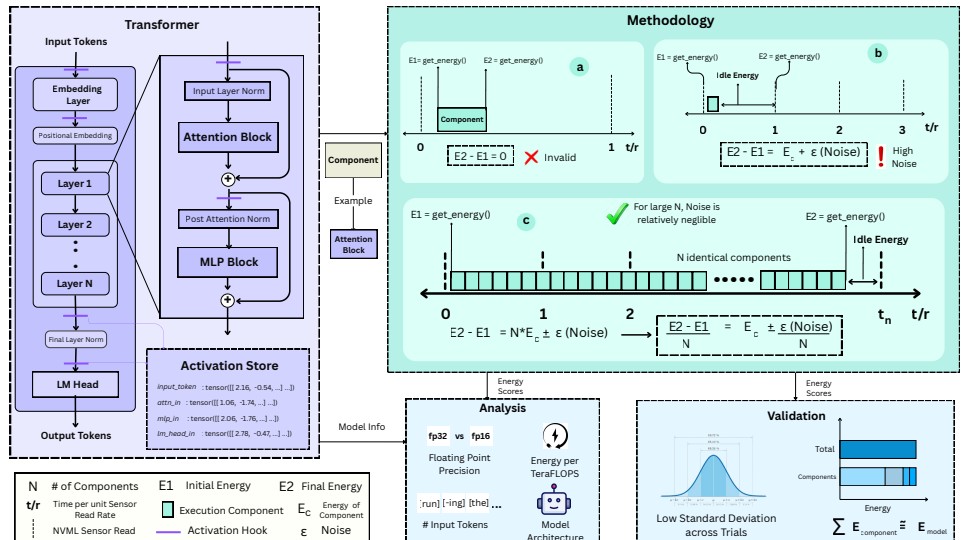

Figure 1: CLEAR pipeline: (1) Store forward-pass activations in Activation Store, (2) Measure per-component energy consumption by isolating each component (e.g., Attention Block) and replaying stored activations, (3) Validate for consistency and completeness, and (4) perform Empirical energy analysis across models.

small scale. Moreover, prior work often relies on FLOPs as a proxy for energy use, despite its limitations. This paper aims to close these gaps by introducing a method to accurately measure and analyze energy consumption at the component level.

## 3 METHODOLOGY

### 3.1 ACTIVATION STORE

Our methodology, CLEAR, targets component-level energy measurement by focusing on key computational primitives common to most transformer-based models as illustrated in Figure 1. These include an **Attention** block that captures token-level dependencies, feed-forward **MLP** blocks for dense nonlinear transformations, normalization blocks (Norm.), the **Embedding Layer** which maps discrete tokens into continuous vector spaces and the final **Language Modeling Head** (LM Head) that project hidden representations back to the vocabulary space for output generation. To enable component-wise energy profiling, we insert forward hooks at key points of the computation graph,

$$\mathcal{A} = \{\texttt{attn\_in}, \texttt{mlp\_in}, \texttt{lm\_head\_in}, \texttt{layer\_norm\_in} \ldots\},$$

and capture the input activations at each hook. During a single forward pass, the hooks record for every component $c \in \mathcal{A}$ the corresponding activation tensor $a_c$ for all tokens,

$$\mathbf{A} = \{a_c \mid c \in \mathcal{A}\}, \qquad a_c \in \mathbb{R}^d.$$

The Activation Store $\mathbf{A}$ (shown in Figure 1) serves as a cache of activations that allows isolated re-execution of individual components under identical input statistics, enabling fine-grained measurement of energy consumption. Refer to Appendix A for details about Transformer architecture.

### 3.2 AMPLIFICATION STRATEGY

Accurately measuring the energy consumption of individual components in a transformer-based LLM is challenging because their execution time is typically much shorter than the sampling period of GPU power sensors. For example, NVIDIA's NVML has a sensor read rate of about 20 to 50 ms. As shown in Methodology of Figure 1, the temporal mismatch leads to two distinct sources of error:

In case (a), when the component completes execution within a few microseconds, entirely between two sensor samples, the monitoring sensor cannot update its reading in time due to which the observed energy is reported as zero:

$$E_2 - E_1 = 0, \tag{1}$$

even though the true component energy is non-zero. This results in a complete underestimation of the component's energy usage.

In case (b), if we supposedly measure energy after every sensor reading to capture the component's energy consumption, the result remains highly noisy. This is because the measurement inevitably includes a significant amount of idle energy drawn by CUDA, making it hard to separate the true component energy. Consequently, when the execution only partially overlaps with a sensor's sampling window, the observed energy is recorded as

$$E_2 - E_1 = E_c \pm \varepsilon, \tag{2}$$

where $E_c$ is the component's actual energy consumed and $\varepsilon$ represents noise.

To address above challenges, we adopt an amplification strategy, illustrated in Methodology (c) of Figure 1. As individual transformer components often complete the execution within 10–100 $\mu s$, their energy consumption remains highly noisy to NVML's coarse sampling window. The goal is to minimize noise ($\varepsilon$) in the component energy measurement to obtain reliable readings. This noise may arise from the model's idle energy consumption or from inherent errors in the sensor measurements. To achieve this, we repeatedly execute each component back-to-back on the cached activations, with no gap between runs. This scales the effective runtime so that the total energy of the repeated executions outweighs the idle background consumption, making the noise comparatively negligible.

Concretely, for each component $c$ with cached input $a_c$, we measure the energy before and after $N$ consecutive executions:

$$E_c^{\text{tot}} = \text{MeasureEnergy}\left( \sum_{i=1}^{N} c(a_c) \right). \tag{3}$$

The per-execution energy can then be obtained by averaging the total measured energy:

$$\hat{E}_c = \frac{E_{\text{end}} - E_{\text{start}}}{N} = \frac{E_c^{\text{tot}}}{N} \pm \frac{\varepsilon}{N}, \tag{4}$$

where $\varepsilon$ denotes the measurement noise. By increasing $N$, the duration of the aggregated workload extends to hundreds of milliseconds which is long enough for NVML's power sensor to capture it while the noise term $\varepsilon/N$ diminishes proportionally, yielding significantly reliable per-component energy estimates. (Refer Algorithm 1)

We repeat the amplified measurement for $T$ trials with a brief pause between runs to let the sensor reset, taking Average and Standard Deviation across trials to further smooth out sensor noise and make the per-component energy estimate more reliable.

---

**Algorithm 1** CLEAR - Component Level Energy Assessment via Repetitions

---

**Require:** Model $\mathcal{M}$, components $\mathcal{C}$, input $x$, iterations $N$, trials $T$
1: Register forward hooks on $\{c \in \mathcal{C}\}$ to capture activations
2: $\mathbf{A} \leftarrow \text{ForwardPassAndCache}(\mathcal{M}, x)$
3: **for** each $c \in \mathcal{C}$ **do**
4:      **for** $t = 1$ to $T$ **do**
5:          $E_c^{\text{tot, (t)}} \leftarrow \text{MeasureEnergy}\left( \sum_{i=1}^{N} c(a_c) \right)$
6:          $\hat{E}_c^{(t)} \leftarrow \frac{E_c^{\text{tot,(t)}}}{N} \pm \frac{\varepsilon}{N}$
7:      **end for**
8:      $\bar{E}_c \leftarrow \frac{1}{T} \sum_{t=1}^{T} \hat{E}_c^{(t)}$
9: **end for**
10: **return** $\{\bar{E}_c \mid c \in \mathcal{C}\}$

---

### 3.3 VALIDATION

Since no existing literature has yet provided the energy consumption of individual components within Transformer architectures, the goal is to validate our methodology along two key dimensions 1) Consistency across trials and 2) Completeness of captured energy

$$\text{StdDev}(\bar{E}_{\text{c}}) \to 0, \qquad \bar{E}_{\text{model}} \approx \sum_{c \in \mathcal{C}} \bar{E}_c.$$

1) A standard deviation close to 0 indicates that repeated component-level energy measurements remain consistent across trials demonstrating high precision in energy measurements by CLEAR. 2) The near-equality between the total measured model energy and sum of its per-component energies demonstrates that CLEAR is able to capture the component's energy usage in a comprehensive manner.

## 4 EXPERIMENTAL DETAILS

### 4.1 HYPERPARAMETER & HARDWARE SPECIFICATIONS

As part of our experimental protocol, we evaluate two floating-point precisions, FP32 and FP16, while varying the input sequence length across 8, 32, 64, 96, and 128 tokens to study scaling effects. Each configuration is run for a fixed set of 20 trials ($T = 20$) to capture variability and validate precision. Assume FP16 precision unless explicitly stated

We also investigate the effect of the repetition count $N$ on reliability of the energy readings. As seen in Fig. 2, increasing repetitions $N$ yields more stable readings and reduces measurement failures i.e. cases where the recorded energy for a trial is spuriously 0 mJ due to sensor granularity. Based on this analysis, we set the repetition count $N = 10,000$ for measurements of small components with execution time of order 100 $\mu$ s and $N = 1,000$ for energy measurements of the full model, balancing measurement accuracy with computational cost.

Experiments were conducted on NVIDIA Ada-Lovelace GPUs (RTX 5000 Ada, RTX 6000 Ada), which incorporate third-generation RT cores and fourth-generation Tensor Cores supporting mixed-precision operations such as FP8 with sparsity NVIDIA (2023); PNY (2023). The RTX 5000 Ada provides 12,800 CUDA cores, 32 GB ECC GDDR6 memory, and a board power of 250 W, while the RTX 6000 Ada offers 18,176 CUDA cores, 48 GB ECC GDDR6 memory, and 300 W board power. However, the NVML interface, which typically updates power readouts only every 20–50 ms (Yang et al., 2024b; Nik et al., 2025), introduced limitations in resolving microsecond-scale component execution, thereby requring the CLEAR methodology for fine-grained energy attribution. Code is available here.

### 4.2 METRICS

#### 4.2.1 ENERGY CONSUMPTION

The energy consumed by each model component is measured in milliJoules(mJ), matching the $\approx$0.8mJ precision of the NVML sensor used. For validating our methodology, we define two complementary metrics, **Energy Captured** (Capture) and **Percentage Capture** (%Capture). Energy Captured (in mJ) represents the total energy measured across all the major components of a given layer block or the entire model. Due to the limited precision of the instrumentation, we neglect negligible contributors (e.g., residual connections) and introduce %Capture to indicate how well the methodology accounts for the model's overall energy usage. Specifically, %Capture is the ratio of the measured Energy Captured to the measured model's energy consumption, expressed as a percentage:

$$\text{Capture} = \sum_{i=1}^{N} \bar{E}_i, \qquad \%\text{Capture} = \frac{\sum_{i=1}^{N} \bar{E}_i}{\bar{E}_{\text{model}}} \times 100$$

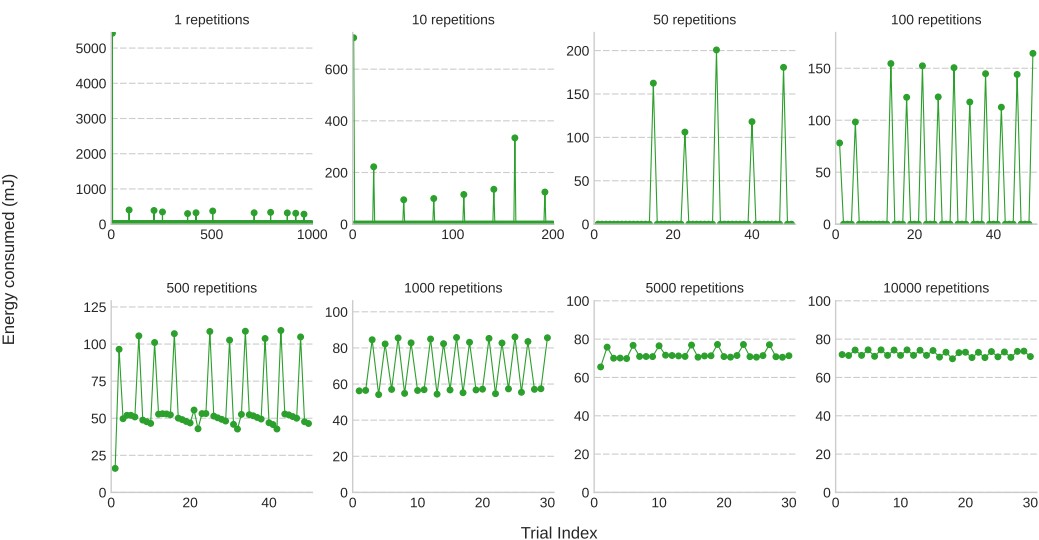

Figure 2: Energy Measurement across multiple trials (T) for varying repetition count N. As N increases, number of reliable readings and precision increases significantly. Y-axis corresponds to Energy Consumption(mJ) and X-axis represents the Trial Index.

### 4.2.2 Computational Efficiency

We use FLOPs to quantify the computations executed by each component for a given input. The PyTorch Profiler is employed to measure both the FLOPs and the GPU execution time (in $\mu$s), providing insights into the computational efficiency of each component. To assess the energy cost per unit of computation, we define two metrics given as

$$
\text{E/FLOP} = \frac{\text{Energy [mJ]}}{\text{FLOPs} \times 10^{-9}}, \qquad \Delta E/\Delta\text{FLOP} = \frac{\Delta Energy \text{ [mJ]}}{\Delta\text{FLOPs} \times 10^{-9}}.
$$

The first metric, E/FLOP (expressed in mJ/GFLOP), characterizes the average energy cost per unit of floating-point computation. Lower values indicate higher energy efficiency, whereas elevated values often signal memory-bound or communication-dominated operations, where energy usage is not directly proportional to computational intensity. The second metric, $\Delta E/\Delta\text{FLOP}$, captures the marginal energy consumed (in mJ) per additional GFLOP and isolates the energy sensitivity of a component to increases in computational demand.

### 4.3 Models

To conduct a systematic study of energy consumption across architectural paradigms, we consider four representative classes of Transformer-based models namely **Encoder- only** models, **Decoder-only** models, **Encoder-Decoder** models and sparse-activated **Mixture of Experts** (MOE) variants. Our model selection aims to balance breadth of architectural diversity with controlled comparisons of scale and design choices.

We evaluate eight widely-used Encoder only models namely BERT-base, BERT-large (Devlin et al., 2019), ALBERT-base, ALBERT-large (Lan et al., 2020), RoBERTa-base, RoBERTa-large (Liu et al., 2019) and distilled variants DistilBERT and DistilBERT (Sanh et al., 2020). Base and large versions allow us to isolate the effect of model size on energy usage where distilled counterparts enable comparison with lightweight compression techniques. To represent contemporary LLMs i.e. Decoder-only models, we experiment with instruction-tuned variants of four key open-source families namely LLaMA 3.2-3B (AI@Meta, 2024), Gemma 3-4B (Team, 2024), Qwen 2.5-3B (Yang et al., 2024a), and Phi-4-4B (Abdin et al., 2024). We focus specifically on single-token generation to control for variability in output sequence length and to minimize cache based auto-regressive generation. We also

| Component | 8 Tokens | | 32 Tokens | | 64 Tokens | | 96 Tokens | | 128 Tokens | |
|---|---|---|---|---|---|---|---|---|---|---|
| | Avg. | Std. Dev. | Avg. | Std. Dev. | Avg. | Std. Dev. | Avg. | Std. Dev. | Avg. | Std. Dev. |
| GPT-OSS 20B | | | | | | | | | | |
| Attention Block | 53.261 | 1.677 | 64.147 | 0.686 | 75.161 | 0.76 | 93.91 | 0.779 | 100.701 | 1.045 |
| MLP | 685.408 | 12.61 | 776.905 | 3.166 | 867.687 | 0.867 | 958.134 | 1.406 | 1046.2 | 1.187 |
| Norm. (All) | 9.324 | 0.729 | 10.787 | 0.825 | 12.702 | 1.056 | 13.443 | 1.422 | 14.639 | 1.108 |
| Captured (Block) | 747.993 | - | 851.839 | - | 955.55 | - | 1065.487 | - | 1161.541 | - |
| Block | 731.905 | 12.456 | 856.309 | 1.428 | 951.869 | 0.805 | 1057.01 | 0.881 | 1157.197 | 1.181 |
| % Capture (Block) | 102.198 | - | 99.478 | - | 100.387 | - | 100.802 | - | 100.375 | - |
| Embedding Layer | 0.568 | 0.215 | 0.627 | 0.282 | 1.061 | 0.41 | 1.077 | 0.434 | 0.766 | 0.357 |
| LM Head | 443.391 | 1.108 | 452.139 | 0.988 | 460.383 | 0.988 | 475.22 | 1.265 | 483.515 | 1.242 |
| Final Layer Norm. | 4.695 | 0.368 | 5.14 | 0.361 | 6.071 | 0.496 | 6.625 | 0.525 | 7.221 | 0.466 |
| Captured (Model) | 18014.38 | - | 21009.32 | - | 23312.36 | - | 25851.15 | - | 28264.24 | - |
| Model | 18447.5 | 63.784 | 21366.69 | 103.479 | 24126.47 | 12.67 | 26634.05 | 15.33 | 28801.98 | 2.867 |
| % Capture (Model) | 97.652 | - | 98.327 | - | 96.626 | - | 97.061 | - | 98.133 | - |

Table 1: Energy Consumption for GPT-OSS-20B model across different token length with %Capture (96+%) for Block and Full Model and Std. Deviation across 20 trials.

evaluate CLEAR on two well-established sequence-to-sequence models, namely BART (Lewis et al., 2019) and FLAN-T5 (Chung et al., 2022) and a sparse-activated MoE, GPT-OSS-20B (OpenAI, 2025)

## 5 RESULTS

### RQ1) HOW CONSISTENT ARE ENERGY MEASUREMENTS ACROSS REPEATED TRIALS, AND HOW COMPLETE IS THE CAPTURED ENERGY RELATIVE TO THE TOTAL CONSUMPTION?

Completeness of Energy Captured: Despite the omission of very small and negligible components, the overall %Capture at both the block and model level consistently remains above 90% (Refer Table 1) This demonstrates that summing the measured energies of individual components provides a reliable and near-complete estimate of the total energy consumption dictated by the model's architecture. However we consistently observe low %Capture(Block) capture of ALBERT variants possible due to factorized embeddings causing higher idle energy consumption.

Consistency Across trials : Using CLEAR, we find that the average standard deviation of the measured component energies consistently remains below 9.5% of the respective mean values for components that consume $> 5$mJ of energy. Furthermore, we observe a clear empirical trend, as both component size and execution time increases, the relative standard deviation decreases. For example, components with energy consumption in the range of 0–5 mJ exhibit around 20+% standard deviation, whereas those consuming 1 J show deviations as low as 0.1 % (Refer Fig 6 in Appendix). Such behavior is expected as shorter execution times result in fewer sensor readings, making the measurements more susceptible to idle-energy noise. Typically, measurements in the 0–5 mJ range are close to the sensor's own precision limit, causing higher % Std. Deviation with respect to the Average.

### RQ2) HOW DOES THE MODEL BEHAVES FROM AN ENERGY PERSPECTIVE UNDER VARYING FLOATING-POINT PRECISIONS?

Empirically we observe that changing the floating-point precision from FP16 to FP32 increases the absolute energy consumption, the relative share of energy consumed by the Attention, MLP, and LM-Head components remains virtually unchanged. Across all model architectures, we observe that normalization layers consume more energy in FP16 precision than in FP32 precision, even under identical settings. This stems from the common practice of casting tensors to 32 bit floating point precision for numerical stability during normalization and then converting back, with the type conversions introducing measurable energy overhead.

For most encoder-only models, the Attention and Feed-Forward blocks consume roughly comparable amounts of energy (Refer Fig 7 and 8 in Appendix). However as model size grows, the dense FFN layers scale up in parameters, so the Attention block accounts for a

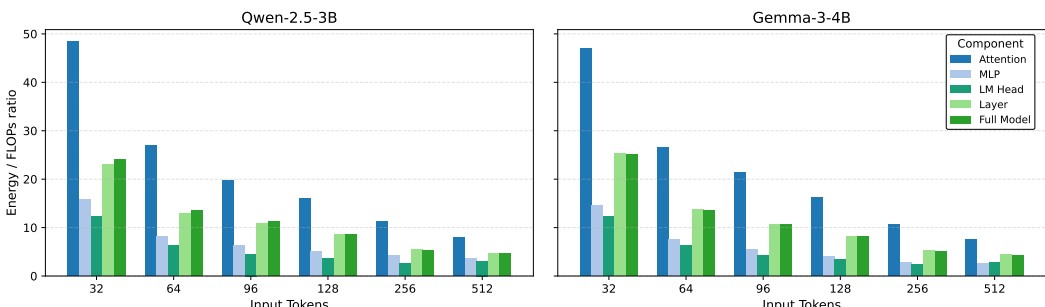

Figure 3: E/FLOP ratio for model components in Qwen-2.5-3B (left) and Gemma-3-4B (right). Attention consistently shows highest E/FLOP ratio across all input lengths.

smaller share of the total energy, mirroring the pattern seen in large decoder-only models (around 3–4 B parameters). (Refer Tables 4, 8, 9, 10 in Appendix)

RQ3) How does the Energy/FLOPs ratio vary across input token length?

As shown in figure 3, we find that across all input token lengths the Attention mechanism consistently exhibits a higher energy-to-FLOPs ratio than an individual layer or the full model as a whole. In contrast, the MLP and LM-head layers consume noticeably less energy per FLOP, which means they perform more computations for every joule of energy spent. This highlights that, from an energy-efficiency perspective, Attention is the most computationally expensive sub-component. This inefficiency stems from the nature of attention computations. Unlike the dense matrix multiplications in the MLP and LM head, which are regular, highly parallelizable operations that GPUs are specifically optimized to accelerate whereas attention involves query–key dot products, scaling, softmax operations, and complex memory access patterns. These steps introduce memory traffic and synchronization overheads, causing the hardware to spend proportionally more energy per unit of floating-point computation.

As shown in figure 4, the energy to FLOPs ratio for every individual component steadily decreases as the input sequence length grows i.e. when we feed the model longer input sequences, each FLOP consumes lesser energy, not only for the complete model, but also for the Attention, MLP, and LM-head blocks. This trend reflects that the fixed costs of computation and memory movement are amortized over more tokens. The per-token energy overheads are diluted and the compute resources are utilized more effectively.

RQ4) Is number of floating-point operations (FLOPs) a reliable indicator of a component's energy consumption?

As shown in Fig. 4, the energy-to-FLOPs ratio (E/FLOPs) decreases consistently as the input sequence length grows. As discussed in **RQ3**, the trend points to the presence of a fixed energy overhead, that is a baseline energy cost $E_0$ likely independent of the number of floating-point operations. Such a cost likely stems from non-scaling aspects of the computation, including memory movements, cache initialization, and kernel-launch overheads, all of which incur an essentially constant energy expenditure regardless of token count.

To quantify the marginal energy required per unit of additional computation, we examined the incremental energy cost per FLOP, $\Delta E/\Delta FLOPs$. Unlike the absolute ratio $E$/FLOPs, the marginal quantity remains approximately constant as the input sequence length increases across all model components. This stability provides strong empirical evidence that FLOPs are indeed the main driver of the **variable** portion of energy consumption. In other words, while total energy can be decomposed as

$$E(L) \approx E_0 + k \cdot \text{FLOPs}(L) \tag{5}$$

with $L$ denoting input length and $k$ a constant of proportionality, only the second term grows nearly linearly with the computational workload (FLOPs).

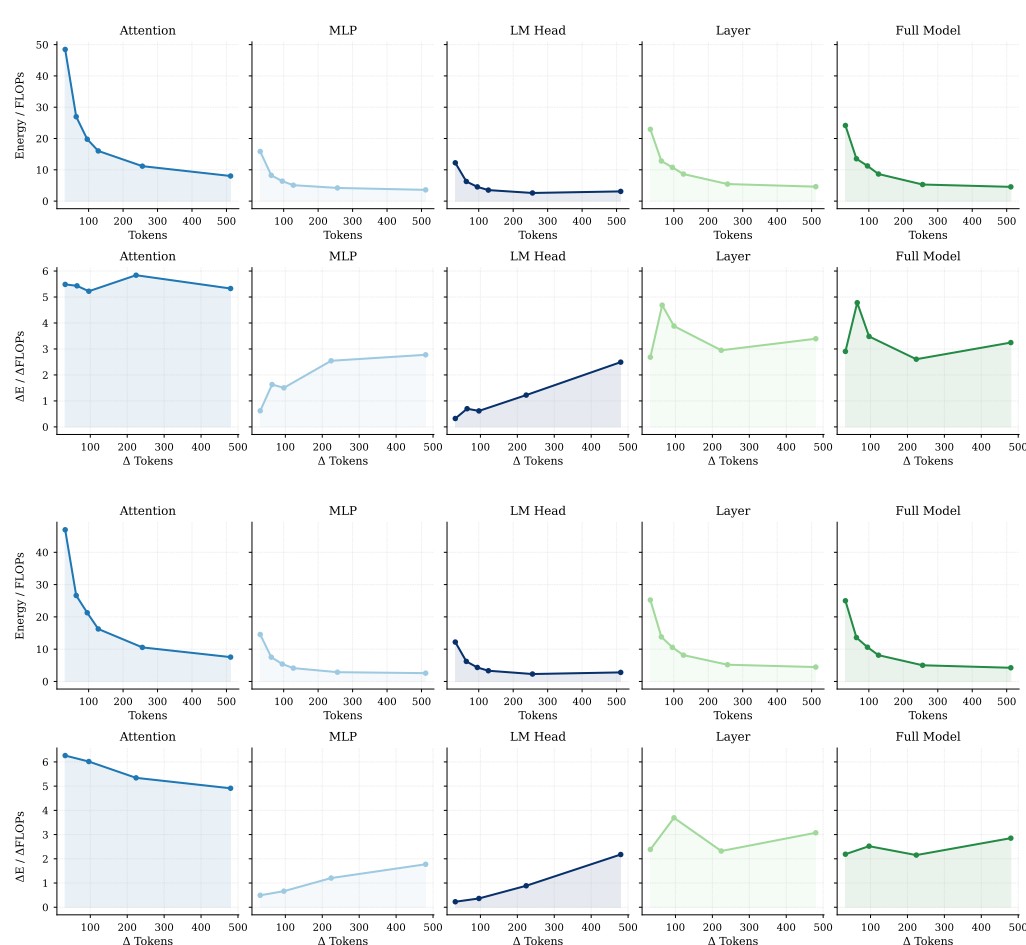

Figure 4: Qwen-2.5-3B: $E$/FLOP(Row-1) and $\Delta E/\Delta$FLOP(Row-2). Gemma-3-4B: $E$/FLOP(Row-3) and $\Delta E/\Delta$FLOP(Row-4). For both models, the $E$/FLOP ratio decreases with input length, while marginal energy per FLOP remains nearly constant.

Notably, the marginal coefficient $k$ is, however, component-dependent. It is noticeably higher for the Attention mechanism than for other subsystems. This can be attributed to the fact that Attention performs repeated key–value cache updates and experiences substantial memory–compute synchronization overheads, both of which increase the effective energy cost per FLOP.

Overall, the empirical observations indicate that energy consumption of different components can be well approximated with two term model comprising of a fixed overhead and a FLOP proportional component. This indicates that, the idea of measuring total energy consumed by a model and then distributing it proportionally across components solely based on FLOPs is overly simplistic. Both the existence of the fixed overhead and the component-specific marginal cost $k$ must be accounted for to estimate energy usage of each component accurately.

## 6 DISCUSSION

Most sustainability studies have primarily focused on model-level or system-level energy consumption treating LLMs as *monolithic entities* and only a little attention is paid to the heterogeneity of it's internal components from a sustainability perspective. CLEAR's contribution to measure energy consumtion at component-level granulity has direct implications

for research community as it provides a systematic methodology to reliably measure internal energy dynamics and enable targeted energy optimizations at the design level of model architecture.

As discussed in **RQ3** and **RQ4**, we observe that each component consumes energy disproportionately, posing a threat to use FLOPs and related metrics as convenient proxies energy consumption (Getzner et al., 2023; Özcan et al., 2025) as component-level disparities are systematically obscured by model-level aggregate measurements. Appendix H for Limitations

As seen in **RQ1**, CLEAR demonstrates statistically reliable and sufficiently complete component-wise energy profiling that can be employedto support comparative studies and draw robust conclusions about energy implications of specific design choices, rather than relying on estimates. CLEAR establishes a foundation for future work on predictive modeling so that the energy costs can be computed based on architectural design choices like hidden dimensions, number of layers, etc. allowing accurate, generalizable prediction of component-wise energy dynamics in the early design stages. This aligns with the growing emphasis on Green AI and the need for energy-aware, sustainable AI system design. (Bolón-Canedo et al., 2024; Sánchez-Mompó et al., 2025b; Różycki et al., 2025)

Taken together, these findings underscore that sustainability in AI must be treated as **a first-class research objective rather than an afterthought**. By moving beyond aggregate model-level reporting to examine component-level dynamics, we aim to motivate the software and AI research communities to pursue progress that is both holistic and environmentally responsible, driven by a proactive rather than reactive mindset. Looking forward, we hope this work inspires future research to integrate energy considerations into every stage of model development, fostering AI systems that are not only performant but also sustainable.

## 7 REPRODUCIBILITY STATEMENT

To ensure transparency and reproducibility, we are committed to making our research accessible. We provide comprehensive experimental details in the paper, and all code will be publicly released upon publication. All experiments were conducted using open-source LLMs.

## 8 ETHICS STATEMENT

This work presents a methodology for measuring energy consumption at the component level in transformer architectures, a dimension that has not been systematically studied so far. Using this methodology, we analyze how energy is distributed across different components of a transformer. Our findings are intended to advance the development of more efficient transformer architectures and reliable methods for energy estimation. Refer I for LLM Usage statement

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

APPENDIX

## A  TRANSFORMER ARCHITECTURES

Despite architectural differences, the transformer based models share a set of common computational primitives. The detailed flow of how the model produces its output is illustrated in the following.

Let $\mathcal{V}$ denote the vocabulary space, and $x = (x_1, \ldots, x_T)$, $x_t \in \mathcal{V}$ be an input token sequence of length $T$. A tokenizer $\mathcal{T} : \mathcal{V} \to \{1, \ldots, |\mathcal{V}|\}$ maps tokens to discrete indices. The indices are embedded into a continuous space by the **Embedding Layer** $E \in \mathbb{R}^{|\mathcal{V}| \times d}$:

$$h_t^0 = E[x_t] + P_t, \quad t = 1, \ldots, T,$$

where $P_t \in \mathbb{R}^d$ denotes the positional embedding and $d$ is the hidden dimensionality. Embeddings are then passed to a set of Transformer layers each consisting of an **Attention** (Attn) block followed by a position-wise Feed-Forward **MLP** block, interleaved with residual connections and **Normalization** blocks for numerical stability, as depicted in the Transformer block of Figure 1. For layer $\ell \in \{1, \ldots, L\}$, the computations are:

$$\tilde{h}^\ell = \mathrm{LN}(h^{\ell-1}), \quad z^\ell = h^{\ell-1} + \mathrm{Attn}(\tilde{h}^\ell), \quad h^\ell = z^\ell + \mathrm{FFN}(\mathrm{LN}(z^\ell)).$$

However, the placement of Normalization blocks vary across different models and can be applied at different stages within a layer block. Post all layers, the final hidden state $h^L \in \mathbb{R}^{T \times d}$ is projected onto the vocabulary using the language modeling head:

$$\hat{y}_t = \mathrm{softmax}(h_t^L W_{\mathrm{LM}}^\top), \quad W_{\mathrm{LM}} \in \mathbb{R}^{|\mathcal{V}| \times d}.$$

### A.1  ENCODER-ONLY

Encoder-only architectures (e.g., BERT, RoBERTa, AlBERT) compute contextualized token representations using bidirectional self-attention. They are commonly used for classification, token-level prediction and masked-language-modeling objectives. The encoder processes the full input $x$ in parallel, producing $H^L = (h_1^L, \ldots, h_T^L) \in \mathbb{R}^{T \times d}$ which can be pooled or projected to task heads.

Encoder-only architectures typically use a *dense attention pattern*, meaning that no causal mask is applied to restrict attention. Formally, the attention operation is defined as

$$\mathrm{Attn}(Q, K, V) = \mathrm{softmax}\left( \frac{QK^\top}{\sqrt{d_k}} \right) V, \tag{6}$$

where every token can attend to every other token in the sequence. This design produces *symmetric contextualization*, since information can flow bidirectionally across tokens. However, it comes with a computational cost of $\mathcal{O}(T^2 d)$ per layer (due to the pairwise interactions between all tokens) and a memory cost of $\mathcal{O}(T^2)$ for storing the attention weights.

Different tasks attach specialized output heads on top of the final hidden states $h_t^L \in \mathbb{R}^d$. For **token-level classification**, each token representation is projected into the label space using a weight matrix $W_{\mathrm{tok}}$. The predicted class distribution for token $t$ is given by

$$\hat{y}_t = \mathrm{softmax}\left( h_t^L W_{\mathrm{tok}}^\top \right).$$

For **sequence-level classification**, the hidden state of a special token such as [CLS] serves as a summary vector for the entire sequence. This representation $h_{\mathrm{cls}}^L$ is then passed through a classifier, typically implemented as a multi-layer perceptron (MLP):

$$h_{\mathrm{cls}}^L \mapsto \mathrm{MLP}(h_{\mathrm{cls}}^L).$$

For **masked language modeling (MLM)**, the prediction head reuses the input embedding matrix $E$ to tie input and output representations. In this case, the output weight matrix is defined as

$$W_{\mathrm{LM}} = E.$$

This weight sharing enforces consistency between how tokens are encoded and how they are predicted.

Encoder-only designs maximize parallelizability during training as the whole sequence is processed concurrently, but the $T^2$ attention cost and the need to store full-layer activations drive both memory bandwidth and energy cost during training.

| Model | # of Layers | Hidden Dimension | Attention Heads | Feed-Forward Dimension | Parameters | Special Features |
|---|---|---|---|---|---|---|
| google-bert/bert-base-uncased | 12 | 768 | 12 | 3072 | 110M | Uses [CLS] token for classification |
| google-bert/bert-large-uncased | 24 | 1024 | 16 | 4096 | 340M | Larger variant of BERT with higher representational capacity |
| albert/albert-base-v2 | 12 | 768 | 12 | 3072 | 12M | Parameter-sharing across layers and factorized embedding |
| albert/albert-large-v2 | 24 | 1024 | 16 | 4096 | 18M | Deeper network with the same parameter-sharing strategy |
| distilbert/distilbert-base-uncased | 6 | 768 | 12 | 3072 | 66M | Distilled version of BERT with 40% fewer parameters |
| distilroberta/distilroberta-base | 6 | 768 | 12 | 3072 | 82M | Distilled version of RoBERTa retaining most performance |
| FacebookAI/roberta-base | 12 | 768 | 12 | 3072 | 125M | Improved pretraining and dynamic masking |
| FacebookAI/roberta-large | 24 | 1024 | 16 | 4096 | 355M | Larger RoBERTa model with improved pretraining |

Table 2: Architectural Details of Encoder-Only Models

## A.2 DECODER-ONLY (AUTOREGRESSIVE)

Decoder-only (autoregressive) models (e.g. Llama3.1, GPT) perform next-token prediction and are optimized for generative tasks. The decoder processes tokens causally with a triangular mask in which each position can attend only to tokens at previous positions (and itself) to enforce autoregressive factorization. The causal mask $M_{\text{causal}}$ has entries 0 for allowed positions and $-\infty$ for disallowed future positions, implementing the triangular attention.

**Inference optimization: KV-caching.** During autoregressive generation, previously computed keys and values can be cached: for step $t$ only the new query interacts with stored $K_{1:t-1}, V_{1:t-1}$. This reduces per-step attention cost from $\mathcal{O}(t^2 d)$ to $\mathcal{O}(td)$ (amortized), and reduces the energy per generated token substantially at inference time. Training a decoder-only model still incurs full-sequence $\mathcal{O}(T^2 d)$ attention cost, but inference benefits from KV-caching. Energy per generated token at inference depends on cache memory bandwidth and per-layer dot-product costs; thus memory movement for KV-cache and tiled attention matmuls can dominate measured energy.

The set of activation hooks $\mathcal{A}_{\text{dec}}$ defined similarly to the encoder case, but is adapted to handle causal inputs and cached key-value states and it stores the intermediate activations $\alpha_{\ell,t}^{\texttt{attn\_in}}$ and $\alpha_{\ell,t}^{\texttt{kv}}$ for each layer $\ell$ and time step $t$. For profiling isolated attention at generation-time, replaying using cached KV tensors models the exact inference cost. For the purpose of our analysis, we primarily consider the energy and computation associated with the generation of **a single token**, where KV-caching is not utilized

Decoder-only models are majorly different from Encoder-only models as they prioritize autoregressive causality in which causal masking changes attention sparsity and reduces parallelism during sequence generation. KV-caching and rotary/relative positional encodings are often used to support long-context amortized inference and decoder-only models commonly use tied input/output embeddings to reduce parameter counts, and favor pre-norm residual stacks for stability in deep networks. Refer Tables 4, 8, 9, 10

## A.3 ENCODER-DECODER

Encoder-Decoder models, also known as sequence-to-sequence (Seq2Seq) architectures, are widely used for tasks requiring input-to-output transformations such as machine translation, summarization, and code generation. Formally, given an input sequence $x = (x_1, \ldots, x_{T_{\text{in}}})$,

| Architecture Detail | Qwen2.5-3B-Instruct | Phi-4-Mini-Instruct | Llama-3.2-3B-Instruct | Gemma-3-4B-IT |
|---|---|---|---|---|
| Parameters | 3.09B total (2.77B non-embedding) | 4B | 3.21B | 4B |
| Layers | 36 | 32 | 28 | 34 |
| Hidden Size / Head Dim | 2048 hidden, 128 per head | 3072 hidden, 128 per head | 3072 hidden, 128 per head | 2560 hidden, 128 per head |
| Attention Structure | GQA: 16 query heads, 2 KV heads; RoPE; QKV bias; output proj. biasless | GQA: 24 query heads, 8 KV heads; Fractional RoPE (25% pos-agnostic); KV cache optimized | GQA: 24 query heads, 8 KV heads; RoPE; no bias in projections | Local+Global attention mix; Q-proj: 2048-d, K/V-proj: 1024-d; q_norm, k_norm applied |
| MLP / FFN Dimension | SwiGLU, 11008 (up+gate), 2048 down | SiLU, 16384 (gate+up), 8192 down | SiLU, 8192 up, 3072 down | GELU-Tanh, 10240 up, 2560 down |
| Normalization | RMSNorm, $\epsilon = 1e{-}6$, applied input + post-attn | RMSNorm, $\epsilon = 1e{-}5$, input + post-attn | RMSNorm, $\epsilon = 1e{-}5$, input + post-attn | RMSNorm, $\epsilon = 1e{-}6$, input + post-attn + pre/post-FFN |
| Embeddings | 151,936 vocab, 2048-d, tied in/out | 200,064 vocab, 3072-d, tied in/out (padding idx=199999) | 128,256 vocab, 3072-d, tied in/out | 262,208 vocab, 2560-d, tied in/out, scaled embeddings |
| Context Length | 32K tokens (gen up to 8K) | Long-context via KV optimization, tested up to ~128K | 128K tokens, efficient GQA | 128K tokens; local layers span 1K, 1 global layer every 5 locals |
| Special Features | RoPE, SwiGLU, QKV bias, high multilingual coverage | GQA w/ reduced KV cache, fractional RoPE, tuned LR schedule | Optimized transformer, SFT+RLHF alignment, multilingual | Local-global hybrid attention, multimodal (SigLIP image encoder), Pan & Scan for variable resolution |

Table 3: Detailed architectural comparison of Decoder-Only Qwen2.5-3B, Phi-4-Mini, Llama-3.2-3B, and Gemma-3-4B instruction-tuned models.

the encoder maps it to a sequence of hidden representations $H = (h_1, \ldots, h_{T_{\mathrm{in}}})$, and the decoder generates an output sequence $y = (y_1, \ldots, y_{T_{\mathrm{out}}})$ autoregressively, conditioned on $H$.

The encoder is a stack of $L_e$ Transformer layers that performs contextual embedding of the input tokens. Each layer typically consists of:

- An **Attention** (Attn) mechanism that captures global dependencies within the input sequence i.e. for layer $\ell$

$$\tilde{h}^{\ell-1} = \mathrm{LN}(h^{\ell-1}), \quad h'^{\ell} = h^{\ell-1} + \mathrm{Attn}^{\ell}(\tilde{h}^{\ell-1}), \quad h^{\ell} = h'^{\ell} + \mathrm{FFN}^{\ell}(\mathrm{LN}(h'^{\ell})).$$

- **Feedforward Network** that adds per-position nonlinear transformation to learn deeper features.

The encoder produces rich representations that capture semantic and syntactic relationships within the input sequence.

The decoder is also a stack of $L_d$ Transformer layers, each consisting of:

- **Masked self-attention** which ensures autoregressive generation by attending only to previous positions.
- **Encoder-decoder cross-attention** mechanism to attend to the encoder hidden states $H$, incorporating information from the entire input sequence into each decoding step.
- A **Feedforward network** similar to the encoder.

Mathematically, for decoder layer $\ell$:

$$\tilde{u}^{\ell-1} = \mathrm{LN}(u^{\ell-1}), \quad s^{\ell}_{\mathrm{self}} = \mathrm{Attn}_{\mathrm{causal}}(\tilde{u}^{\ell-1}),$$

$$s^{\ell}_{\mathrm{cross}} = \mathrm{Attn}_{\mathrm{cross}}(\mathrm{LN}(u^{\ell-1} + s^{\ell}_{\mathrm{self}}), H), \quad u^{\ell} = \mathrm{FFN}(\mathrm{LN}(u^{\ell-1} + s^{\ell}_{\mathrm{self}} + s^{\ell}_{\mathrm{cross}})).$$

Compared to encoder-only models, the encoder-decoder architecture introduces a separate decoder stack with cross-attention, which enables output generation conditioned on the full input sequence. In contrast, encoder-only models produce fixed-length or token-level representations that are typically used for classification or embedding tasks, without any autoregressive generation.

When compared to decoder-only models, encoder-decoder architectures separate the input encoding from the output generation, whereas decoder-only models combine both within a single autoregressive stack. This separation allows the encoder to process the entire input sequence in parallel, improving training efficiency. Furthermore, in terms of residual and attention patterns, encoder-decoder models incorporate both self-attention in the decoder and cross-attention between the decoder and encoder outputs, whereas encoder-only and decoder-only architectures contain only a single attention mechanism.

**Energy Perspective:** The two-stack design of encoder-decoder models increases the total parameter count and memory footprint, resulting in higher energy consumption during training compared to encoder-only models for sequences of the same length. However, the input encoding phase can be fully parallelized across positions, and autoregressive decoder computation can benefit from caching mechanisms during inference, which partially reduces the per-token energy cost

### A.4 MIXTURE OF EXPERTS (MoE)

Mixture-of-Experts (MoE) architectures extend standard Transformers by introducing conditional computation i.e. instead of activating all parameters for every input token, only a subset of "expert" networks is selected dynamically. This allows scaling model capacity substantially while keeping per-token computation and energy consumption manageable.

An MoE layer contains $E$ independent feedforward networks, or Experts, each with parameters $\theta_1, \ldots, \theta_E$. For a given token representation $h \in \mathbb{R}^d$, the computation is routed through a small subset of $k < E$ experts, typically $k = 2$ or $3$

$$m^{\text{MoE}}(h) = \sum_{i \in \text{Top-}k} g_i(h) \, \text{FFN}_i(h),$$

where $g_i(h)$ is the gating weight assigned by the Router. By activating only a few experts per token, the effective FLOPs per token can be reduced from $\mathcal{O}(E \cdot d \cdot d_{\text{ff}})$ to $\mathcal{O}(k \cdot d \cdot d_{\text{ff}})$.

The Router is a lightweight module that predicts which experts should process a given token:

$$g(h) = \text{softmax}(hW_r), \qquad g \in \mathbb{R}^E.$$

It selects the top-$k$ experts according to the largest $g_i$ values. The Router can also include auxiliary losses, such as load-balancing or importance losses, to encourage uniform expert utilization and avoid stragglers, which would increase memory or energy spikes.

By increasing the total number of experts $E$ without increasing $k$, it is possible to scale the model's representational capacity while incurring only a small incremental energy cost per token. In practice, expert computations for different tokens are often batched across devices, but load imbalance can increase memory movement and create temporary energy spikes due to which careful load-balancing and token assignment become necessary to maintain efficiency.

## B HARDWARE SPECIFICATION

The experiments in this paper were carried out using NVIDIA's Ada-Lovelace architecture GPUs, namely the RTX 5000 Ada and RTX 6000 Ada, in order to assess compute and energy performance. The Ada Lovelace architecture is fabricated on a custom 4 nm TSMC process and includes third-generation RT cores and fourth-generation Tensor cores, enabling mixed precision operations (including FP8 with sparsity) that are integral to efficient transformer inference NVIDIA (2023). According to the official datasheets, the RTX 5000 Ada has 12,800 CUDA cores, 100 RT cores, 400 Tensor cores, 32 GB of ECC GDDR6 memory over a 256-bit interface (providing ∼576 GB/s bandwidth), and a total board power of approximately 250 W PNY (2023). The RTX 6000 Ada model offers 18,176 CUDA cores, 142 RT cores, 568 Tensor cores, 48 GB of ECC GDDR6 memory on a 384-bit interface (∼960 GB/s bandwidth), and has a board power of around 300 W NVIDIA (2023). These hardware choices directly

influence both the sustained compute throughput and the energy-per-FLOP metrics reported in our results.

NVIDIA does not publish the precise NVML power sampling interval for the RTX 5000 Ada or RTX 6000 Ada. Prior work has shown that on modern NVIDIA GPUs, NVML's power readouts are typically updated at a frequency of 20–50 Hz (i.e., every 20–50 ms), which constrains the granularity of fine-grained energy attribution Yang et al. (2024b), Nik et al. (2025).

| Components | Qwen2.5-3B fp32 | | Qwen2.5-3B fp16 | | Llama-3.2-3B fp32 | | Llama-3.2-3B fp16 | | Gemma-3-4B fp32 | | Gemma-3-4B fp16 | |
|---|---|---|---|---|---|---|---|---|---|---|---|---|
| | Mean | Std. Dev | Mean | Std. Dev | Mean | Std. Dev | Mean | Std. Dev | Mean | Std. Dev | Mean | Std. Dev |
| MLP | 113.71 | 6.687 | 48.5 | 2.13 | 127.24 | 1.94 | 54.55 | 2.01 | 129.47 | 1.86 | 60.07 | 2.13 |
| Attention | 27.64 | 1.79 | 33.99 | 5.96 | 58.31 | 3.01 | 23.88 | 1.42 | 42.82 | 2.11 | 36.84 | 1.85 |
| Input Layer Norm | 2.59 | 0.24 | 6.42 | 1.02 | 3.2 | 0.25 | 3.61 | 0.41 | 3.74 | 0.45 | 4.42 | 0.43 |
| Attention Layer Norm | 2.81 | 0.33 | 6.8 | 0.97 | 3.2 | 0.32 | 3.85 | 0.41 | 3.49 | 0.33 | 4.52 | 0.36 |
| Capture (Block) | 146.75 | - | 95.71 | - | 191.95 | - | 85.89 | - | 186.54 | - | 114.83 | - |
| Block | 150.03 | 3.39 | 96.19 | 5.15 | 192.26 | 24.92 | 91.85 | 1.98 | 187.54 | 3.91 | 126.63 | 4.54 |
| %Capture (Block) | 97.81 | | 99.50 | | 99.84 | | 93.51 | | 99.47 | | 90.68 | |
| Final Layer Norm | 2.96 | 0.41 | 6.94 | 0.97 | 3.2 | 0.27 | 4.2 | 0.58 | 3.5 | 0.33 | 4.53 | 0.32 |
| Embedding | 0.81 | 0.26 | 0.74 | 0.06 | 0.69 | 0.23 | 0.68 | 0.24 | 1.52 | 0.29 | 1.28 | 0.27 |
| LLM Head | 459.66 | 2.18 | 214.29 | 3.56 | 602.02 | 2.97 | 374.92 | 2.96 | 1040.63 | 7.25 | 480.64 | 8.72 |
| Model | 5864.51 | - | 3684.81 | - | 5989.19 | - | 2951.6 | - | 7422.01 | - | 4791.87 | - |
| Capture (Model) | 5995.27 | 26.77 | 3685.95 | 29.11 | 6029.32 | 10.68 | 3261.5 | 30.99 | 8086.96 | 25.25 | 5248.99 | 89.34 |
| %Capture (Model) | 97.82 | | 99.97 | | 99.33 | | 90.50 | | 91.78 | | 91.29 | |

Table 4: CLEAR demonstrating similar performance on RTX 5000 GPU for Decoder-only models(Qwen2.5-3B, Llama-3.2-3B, Gemma-3-4B) across fp16 and fp32 floating point precisions.

# C  NVIDIA RTX 5000 AND RTX 6000

We validate our methodology across three models on the NVIDIA RTX 5000 ADA GPU and observe a %Capture exceeding 90%, with minimal standard deviation across both fp16 and fp32 precisions. Interestingly, the energy consumption of normalization blocks remains higher for fp16 compared to fp32, similar to the trend observed on the NVIDIA RTX 6000. Refer Tables 4, 8, 9, 10

# D  ATTENTION VARIANTS & OPTIMIZATIONS

Using CLEAR, we compare three Attention implementations, namely *Eager Attention, Scaled Dot-Product Attention (SDPA)*, and *Flash Attention.* All Attention variants are computed using equation 6 , but they differ in the way operations are executed on the GPU (hardware layer).

Eager Attention uses separate kernels for each step, which increases memory transfers and induces a launch-time overhead. SDPA reduces launch-time overhead by fusing some operations, but it still creates the full attention matrix. Flash Attention goes further by computing attention in tiled blocks without storing the entire matrix, which reduces memory usage and improves efficiency.

We also evaluate three optimization strategies. **Torch compile** performs whole graph optimizations and merges many small GPU kernels into larger fused kernels. **Max Autotune** spends additional compilation time to recognize the best kernel implementations for the current code execution. It runs many candidate kernels and selects the best-performing one for the current hardware setup. Optimization using Max Autotune is achieved mainly due to Aggressive kernel fusion which involves combining multiple operations into single kernels and Autotuning that tries out different tile sizes, block sizes, memory layouts for given hardware. **Reduced Overhead** removes profiling and extra synchronization steps to simplify execution. It focuses on reducing kernel launch overhead and minimizes the time spent switching between CPU and GPU.

| Model & Input Length | Attention Setup | Energy Consumption (mJ) | Std. Deviation | FLOPs (in GFLOPs) | Energy (mJ) / GFLOP |
|---|---|---|---|---|---|
| Qwen2.5 - 3B
64 input tokens | FP16 Flash Attention | 38.609 | 0.348 | 1.208 | 31.950 |
| | FP16 Eager Attention | 45.398 | 0.564 | 1.242 | 36.552 |
| | FP16 SDPA | 35.061 | 0.378 | 1.208 | 29.014 |
| | FP16 Flash with torch.compile() | 29.100 | 0.320 | 1.208 | 24.090 |
| | FP16 Eager with torch.compile() | 31.675 | 0.163 | 1.242 | 25.513 |
| | BF16 Flash Attention | 48.289 | 1.247 | 2.417 | 19.980 |
| | FP16 Eager Max Autotune | 19.679 | 0.664 | - | - |
| | F16 Eager Reduce Overhead | 19.640 | 0.765 | - | - |
| Qwen2.5 - 3B
128 input tokens | FP16 Flash Attention | 44.169 | 0.764 | 2.417 | 18.276 |
| | FP16 Eager Attention | 53.766 | 1.506 | 2.551 | 21.074 |
| | FP16 SDPA | 39.577 | 0.642 | 2.417 | 16.376 |
| | FP16 Flash with torch.compile() | 33.337 | 0.441 | 2.416 | 13.799 |
| | FP16 Eager with torch.compile() | 38.881 | 0.947 | 2.550 | 15.247 |
| | BF16 Flash Attention | 48.289 | 1.247 | 2.417 | 19.980 |
| | FP16 Eager Max Autotune | 24.977 | 0.318 | - | - |
| | F16 Eager Reduce Overhead | 24.233 | 0.881 | - | - |
| Qwen2.5 - 3B
256 input tokens | FP16 Flash Attention | 56.665 | 1.131 | 4.834 | 11.723 |
| | FP16 Eager Attention | 70.411 | 0.585 | 5.372 | 13.108 |
| | FP16 SDPA | 62.223 | 1.319 | 4.834 | 12.873 |
| | FP16 Flash with torch.compile() | 47.558 | 0.702 | 4.832 | 9.843 |
| | FP16 Eager with torch.compile() | 59.949 | 0.818 | 5.369 | 11.166 |
| | BF16 Flash Attention | 48.289 | 1.247 | 2.417 | 19.980 |
| | FP16 Eager Max Autotune | 33.500 | 0.874 | - | - |
| | F16 Eager Reduce Overhead | 36.626 | 1.095 | - | - |
| Gemma3 - 4B
64 input tokens | FP16 Flash Attention | 59.416 | 0.882 | 2.014 | 29.498 |
| | FP16 Eager Attention | 66.962 | 0.694 | 2.048 | 32.699 |
| | FP16 SDPA | 57.048 | 1.035 | 2.014 | 28.322 |
| | FP16 Flash with torch.compile() | 32.953 | 0.461 | 2.013 | 16.368 |
| | FP16 Eager with torch.compile() | 34.322 | 0.383 | 2.047 | 16.769 |
| | BF16 Flash Attention | 67.633 | 0.986 | 2.014 | 33.577 |
| | FP16 Eager Max Autotune | 23.842 | 0.762 | - | - |
| | F16 Eager Reduce Overhead | 23.827 | 0.841 | - | - |
| Gemma3 - 4B
128 input tokens | FP16 Flash Attention | 69.228 | 0.749 | 4.029 | 17.184 |
| | FP16 Eager Attention | 76.126 | 0.658 | 4.163 | 18.287 |
| | FP16 SDPA | 65.785 | 0.504 | 4.029 | 16.330 |
| | FP16 Flash with torch.compile() | 43.885 | 0.558 | 4.027 | 10.899 |
| | FP16 Eager with torch.compile() | 44.267 | 0.562 | 4.161 | 10.639 |
| | BF16 Flash Attention | 76.328 | 1.326 | 4.029 | 18.947 |
| | FP16 Eager Max Autotune | 30.591 | 0.852 | - | - |
| | F16 Eager Reduce Overhead | 31.651 | 0.851 | - | - |
| Gemma3 - 4B
256 input tokens | FP16 Flash Attention | 85.843 | 1.854 | 8.057 | 10.654 |
| | FP16 Eager Attention | 100.152 | 2.179 | 8.594 | 11.653 |
| | FP16 SDPA | 93.987 | 1.188 | 8.057 | 11.665 |
| | FP16 Flash with torch.compile() | 58.833 | 0.814 | 8.053 | 7.306 |
| | FP16 Eager with torch.compile() | 60.907 | 0.696 | 8.590 | 7.091 |
| | BF16 Flash Attention | 101.571 | 3.304 | 8.057 | 12.607 |
| | FP16 Eager Max Autotune | 41.614 | 0.619 | - | - |
| | F16 Eager Reduce Overhead | 44.674 | 0.161 | - | - |

Table 5: Average Energy Consumption, FLOPs and Energy(mJ)/GFLOPs ratio for Gemma3-4B and Qwen2.5-3B models across input token lengths of 64, 128 and 256. We demonstrate results for 3 Attention Variants (SDPA, Eager, Flash) along with Optimizations such as Torch Compile, Max Autotune and Reduced Overhead

Results in Table 5 show that energy consumption does not increase in direct proportion to FLOPs. Instead, the measurements match the relationship given by the equation 5 Here, $E_0$ includes kernel launch overhead, memory allocation, and setup costs. Because of the fixed cost incurred due to launch overhead and memory allocation, the energy per GFLOP decreases as the sequence length increases. In some hardware architecture, BF16 math units are internally convert to FP32, while FP16 has more native support. These additional conversion steps introduce marginally higher energy cost for BF16 as compared to FP16 precision.

As shown in results of Table 5, across all input token lengths, Flash Attention uses energy comparable to SDPA but much lower than naive Eager Attention. Modern GPUs spend a large portion of energy on memory access and kernel launches, and not just arithmetic computations. Flash Attention reduces the memory traffic significantly and when combined with Torch Compile, the fused kernels help to further reduce the energy and latency and increase efficiency. Torch Compile allows for optimization of the Eager Attention mechanism by merging many smaller GPU kernels into larger ones.

Optimizations such as Max Autotune and Reduced Overhead show a drastic drop in overall energy consumption. Such optimizations use fused execution paths that have fewer intermediate operations helping to reduce computation and energy movement.

For the Max Autotune and Reduced Overhead settings, it is not possible to measure FLOPs using standard profiling tools. As the above optimizations create execution graphs that do not match the usual operator-level structure, many operations are fused or replaced by hardware-specific kernels chosen at runtime. As the fused kernels do not correspond to individual matrix multiplications or softmax operations, tools cannot assign a reliable FLOP count.

## E    MULTI-TOKEN GENERATION

Using CLEAR's methodology, we extend our evaluation beyond the single-token Prefill stage to a more realistic multi-token generation for decoder-only transformer models. Unlike single-token experiments multi-token generation encompasses both the Prefill phase and the Decode phase but introduces new computational challenges due to Key–Value (KV) cache optimizations.

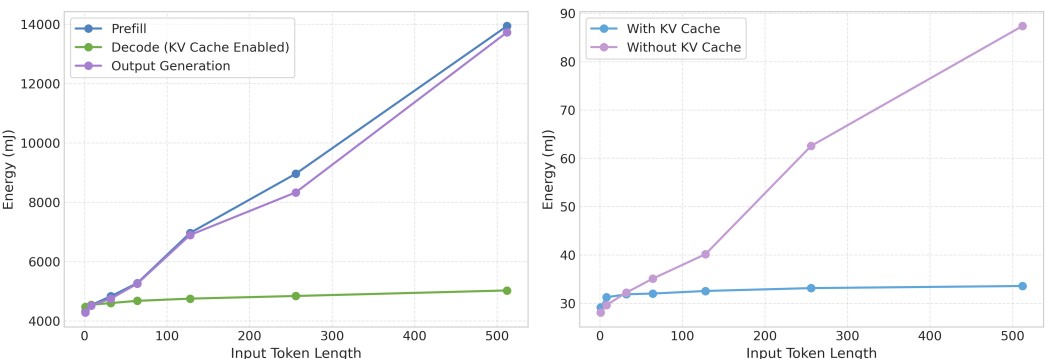

Figure 5: Left: Energy Scaling of Prefill, Decode Stage (with KV Cache Enabled) and Output Generation with input token lenghts for Qwen2.5-3B model (FP16). Right: Energy Scaling of Attention Mechanism with and without KV Cache for Qwen2.5-3B model (FP16)

In decoder-based models, autoregressive generation proceeds a single token at a time. For each new token, the model requires to compute Attention over entire input and previously generated tokens. Without caching, it would require re-computing Key and Value projections for the entire sequence at every step, incurring quadratic scaling in both FLOPs and energy. KV Cache eliminates this computational redundancy by storing previously computed K and V tensors and reusing them in subsequent decode steps.

| Input Token Length | With KV Cache | | | Without KV Cache | | |
|---|---|---|---|---|---|---|
| | Avg Energy (mJ) | Std Dev. | GFLOPS | Avg Energy (mJ) | Std Dev. | GFLOPS |
| 1 | 29.118 | 0.515 | 0.019 | 28.052 | 0.474 | 0.019 |
| 8 | 31.217 | 0.524 | 0.019 | 29.622 | 0.682 | 0.151 |
| 32 | 31.850 | 0.511 | 0.019 | 32.184 | 0.580 | 0.604 |
| 64 | 31.995 | 0.513 | 0.019 | 35.082 | 0.690 | 1.208 |
| 128 | 32.532 | 0.239 | 0.019 | 40.148 | 0.696 | 2.417 |
| 256 | 33.124 | 0.785 | 0.019 | 62.554 | 1.591 | 4.834 |
| 512 | 33.558 | 1.335 | 0.019 | 87.369 | 2.220 | 9.667 |

Table 6: Energy consumption and FLOP requirements for the attention mechanism in the Qwen2.5-3B model with and without KV cache across varying input sequence lengths. When KV cache is enabled, both energy usage and computational cost remain nearly constant, whereas disabling the KV cache leads to a sharp increase in FLOPs and energy as sequence length grows.

As shown in Table 7, multi-token generation exhibits markedly different scaling characteristics across the Prefill phase, Decode phase, and complete Output Generation pipeline for a single new token. The Prefill phase processes the complete input sequence and shows a near-linear increase in both FLOPs and Energy Consumption as input length grows, consistent with the need to compute fresh Q,K and V projections for all tokens. In contrast, the Decode stage remains almost invariant to sequence length, maintaining constant FLOPs ($\tilde{6}.17$ GFLOPs) and only a marginal rise in energy due to KV Cache reuse for larger sequences. KV Cache eliminates the redundant computation by retrieving previously computed keys and values.

As shown in Figure 5, entire `model.generate()` pipeline scales similar to the Prefill stage because computation of logits and framework-level overheads still depend on sequence length and partially bypass cache optimizations. Collectively, the observations underscore that KV Cache is a dominant factor to achieve long-context decoding, while Prefill and Output Generation pipelines remain the primary contributors to energy growth in realistic generation scenarios.

Table 6 reveals that for the Attention Mechanism, enabling the KV cache keeps both FLOPs and Energy Consumption nearly constant, as the model only needs to process newly generated token and can reuse all previously stored keys and values. In contrast, when the KV cache is disabled, the computational costs rises sharply as the input sequence grows. Such behavior arises because the Attention Mechanism needs to recompute all Q, K, V The 'CLEAR' gap between the two behaviors demonstrates that KV Cache effectively eliminates redundant computation and is essential for efficient, long-context Autoregressive Decoders.

## F  VARIATION IN STANDARD DEVIATION

As shown in Fig. 6, the standard deviation of energy measurements exhibits a higher relative deviation at lower energy values (around 1 mJ), primarily due to the limited precision of the NVML energy sensor. For measurements above 5 mJ, the deviation stabilizes to an acceptable range of approximately 9%, and further decreases below 5% for energies exceeding 20 mJ. This behavior arises because fixed sensor resolution introduces proportionally larger errors at smaller measurement scales.

## G  DISTRIBUTION OF ENERGY ACROSS COMPONENTS

We analyse 12 models using pie charts to show the Distribution of energy across components. We empirically observe the following:

| Input Token Length | Prefill | | Next token Decode Stage | | Output Generation | |
|---|---|---|---|---|---|---|
| | Avg Energy (mJ) | GFLOPs | Avg Energy (mJ) | GFLOPs | Avg Energy (mJ) | GFLOPs |
| 1 | 4329.42 | 6.17 | 4472.20 | 6.17 | 4278.49 | 6.17 |
| 8 | 4531.80 | 49.38 | 4543.73 | 6.17 | 4517.82 | 45.02 |
| 32 | 4829.55 | 197.51 | 4600.39 | 6.17 | 4749.55 | 178.22 |
| 64 | 5273.63 | 395.03 | 4675.02 | 6.17 | 5254.10 | 355.82 |
| 128 | 6964.34 | 790.06 | 4749.80 | 6.17 | 6895.61 | 711.02 |
| 256 | 8959.38 | 1580.12 | 4839.23 | 6.17 | 8330.42 | 1421.42 |
| 512 | 13940.02 | 3160.24 | 5025.20 | 6.17 | 13730.38 | 2842.23 |

Table 7: Energy consumption and FLOP analysis of the Qwen2.5-3B model across varying input sequence lengths. We report measurements for three important phenomena: Prefill, Next-Token Decode (KV Cache Enabled) and Output Generation pipeline with 1 new token. The results show that while the FLOPs of the decode step remain constant due to KV reuse, its energy consumption increases only marginally with longer input lengths, in contrast to the Prefill and Output Generation where energy consumption scales rapidly with the input sequence length.

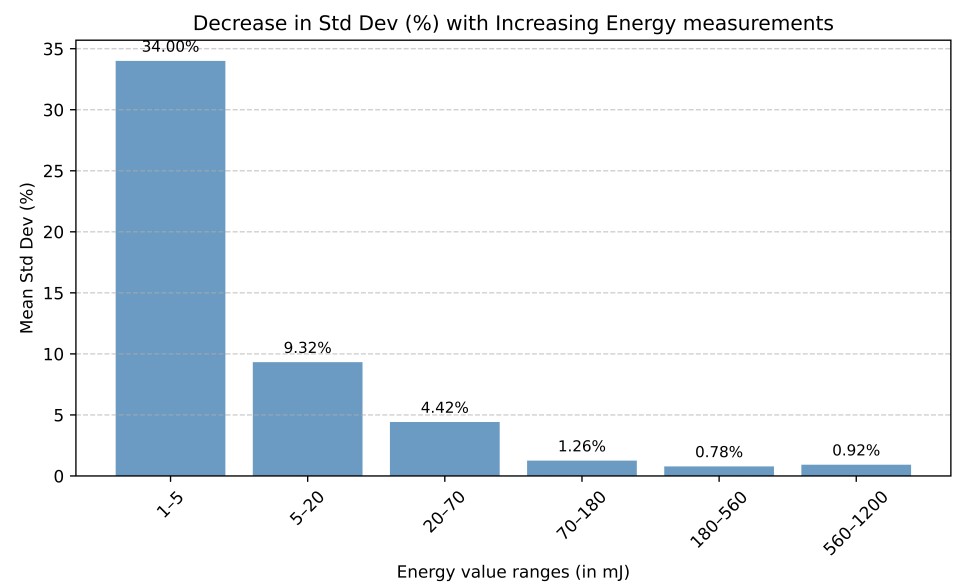

Figure 6: Standard Deviation across different Energy measurements. We observe a decrease in Standard Deviation in our Energy Measurement Approach with increasing Energy Values.

In encoder-only models (Refer Fig.7), the Attention block accounts for a substantial share of energy consumption (typically $> 40\%$). However, in their larger variants (Refer Fig.8), the share of energy attributed to MLP layers increases markedly. For decoder-only models (Refer Fig.9), MLP layers consistently dominate, consuming more than $50\%$ of the total energy. The trend highlights that MLP energy usage is strongly tied to hidden dimension size and the presence of projection layers within the block. In very large models (e.g., GPT-OSS-20B), the Attention contribution becomes negligible relative to the MLP, driven by the dense feedforward computations that scale with model size.

Across all models, normalization and embedding layers consume $< 5\%$ each, reinforcing that the dominant energy trade-off is between MLP and Attention. The energy consumed by

the LM Head is largely dependent on the Vocabulary space $|V|$ of the model and increases significantly in case of Gemma-3-4B and Phi-4-4B models as compared other decoder-only models.

Across all models, the relative shares of MLP, Attention, and LM Head remain stable whether FP16 or FP32 is used. We can thus observe that precision scaling reduces absolute energy but preserves the internal proportion, keeping MLP the dominant bottleneck regardless of precision.

## H   LIMITATIONS

While our study provides the first component-level view of energy consumption in Transformer architectures, a few aspects merit further exploration. First, our energy estimates rely on NVIDIA's NVML interface and FLOP counts obtained via the PyTorch Profiler. These are well-established tools, but like all measurement frameworks they may carry some inherent uncertainties that may slightly affect the results by negligible margin.

Second, GPU hardware introduces additional variability. Different GPU families and generations apply their own low-level optimization, potentially affecting the energy profiles of specific computational components. Extending this analysis across a wider range of hardware would sharpen our understanding of how these optimizations influence component-wise energy usage. While this is left for future work, to the best of our knowledge, our study remains the first systematic investigation of component-level energy dynamics in Transformer models, providing a foundational understanding of the challenges and opportunities present on current hardware.

Finally, because prior literature offers little precedent for fine-grained energy measurement of individual Transformer components, our validation protocol represents an initial step. Future work can strengthen and expand these validation methods as the research community develops more sophisticated benchmarks and measurement standards.

## I   LLM USAGE

We used LLMs as writing assistants to help polish and clarify the text in this paper motivated to improve clarity and consistency.

| Component | 32 Tokens | | 64 Tokens | | 96 Tokens | | 128 Tokens | |
|---|---|---|---|---|---|---|---|---|
| | Avg. | Std. Dev. | Avg. | Std. Dev. | Avg. | Std. Dev. | Avg. | Std. Dev. |
| Gemma-3 4B | | | | | | | | |
| Attention Block | 47.312 | 1.421 | 53.619 | 1.672 | 64.300 | 1.381 | 65.488 | 1.050 |
| MLP | 73.218 | 1.043 | 75.700 | 0.872 | 81.383 | 0.776 | 83.230 | 0.995 |
| Norm. (All) | 26.144 | 0.533 | 27.853 | 0.456 | 30.892 | 0.394 | 32.878 | 0.877 |
| Captured (Block) | 146.674 | - | 157.173 | - | 176.575 | - | 181.596 | - |
| Block | 152.390 | 2.879 | 166.800 | 3.389 | 191.299 | 3.063 | 197.011 | 3.642 |
| % Capture (Block) | 96.249 | - | 94.229 | - | 92.303 | - | 92.176 | - |
| Embedding Layer | 1.916 | 0.399 | 2.078 | 0.430 | 2.147 | 0.451 | 2.258 | 0.454 |
| LM Head | 524.027 | 1.673 | 533.766 | 1.431 | 558.508 | 1.785 | 570.688 | 1.820 |
| Final Layer Norm. | 6.318 | 0.348 | 6.644 | 0.419 | 7.399 | 0.394 | 7.871 | 0.284 |
| Captured (Model) | 5713.511 | - | 6213.671 | - | 7072.219 | - | 7279.177 | - |
| Model | 6208.532 | 116.391 | 6752.334 | 88.871 | 7803.634 | 114.325 | 8047.096 | 104.472 |
| % Capture (Model) | 92.027 | - | 92.023 | - | 90.627 | - | 90.457 | - |
| Qwen2.5 3B | | | | | | | | |
| Attention Block | 29.281 | 1.056 | 32.595 | 1.417 | 58.270 | 1.295 | 67.588 | 1.540 |
| MLP | 68.649 | 1.394 | 71.329 | 1.926 | 163.531 | 0.565 | 167.899 | 0.807 |
| Norm | 10.560 | 2.141 | 11.796 | 1.939 | 21.905 | 2.161 | 22.756 | 2.172 |
| Captured (Block) | 108.490 | - | 115.721 | - | 243.706 | | 258.242 | |
| Block | 113.174 | 1.417 | 126.411 | 2.573 | 246.125 | 5.963 | 260.012 | 3.274 |
| % Capture (Block) | 95.861 | - | 91.543 | - | 99.017 | | 99.319 | |
| Embedding Layer | 1.047 | 0.407 | 2.514 | 0.937 | 2.648 | 1.108 | 2.794 | 1.014 |
| LM Head | 243.657 | 3.644 | 250.085 | 0.974 | 530.914 | 2.714 | 544.742 | 2.560 |
| Final Layer Norm | 5.149 | 1.057 | 5.451 | 0.426 | 10.769 | 1.012 | 11.206 | 1.117 |
| Captured (Model) | 4324.124 | - | 4808.860 | - | 9404.846 | - | 9919.159 | - |
| Model | 4772.004 | 105.363 | 5345.951 | 133.036 | 10322.788 | 66.151 | 10727.263 | 84.912 |
| % Capture (Model) | 90.614 | - | 89.953 | - | 91.108 | - | 92.467 | - |
| Phi3 4B | | | | | | | | |
| Attention Block | 67.834 | 1.684 | 72.138 | 1.266 | 81.806 | 1.367 | 89.256 | 1.127 |
| MLP | 134.556 | 1.434 | 148.118 | 1.147 | 155.105 | 1.371 | 168.813 | 0.799 |
| Norm | 20.899 | 2.071 | 21.425 | 2.088 | 22.952 | 2.187 | 23.385 | 2.118 |
| Captured (Block) | 223.289 | - | 241.681 | - | 259.864 | - | 281.453 | - |
| Block | 229.321 | 1.148 | 264.038 | 1.582 | 281.425 | 2.429 | 298.016 | 2.439 |
| % Capture (Block) | 97.370 | - | 91.533 | - | 92.338 | - | 94.442 | - |
| Embedding Layer | 2.773 | 1.058 | 2.731 | 1.043 | 2.769 | 1.089 | 2.948 | 1.142 |
| LM Head | 957.378 | 4.961 | 995.880 | 13.324 | 1057.904 | 12.474 | 1073.472 | 13.378 |
| Final Layer Norm | 10.237 | 0.979 | 10.525 | 1.048 | 11.284 | 1.109 | 11.640 | 1.096 |
| Captured (Model) | 8308.650 | - | 9458.356 | - | 10077.566 | - | 10624.575 | - |
| Model | 9204.600 | 57.325 | 10331.063 | 92.475 | 11397.891 | 93.548 | 11833.962 | 135.785 |
| % Capture (Model) | 90.266 | - | 91.553 | - | 88.416 | - | 89.780 | - |
| Llama3.2-3B | | | | | | | | |
| Attention Block | 45.932 | 0.763 | 54.380 | 0.706 | 61.239 | 0.839 | 66.667 | 0.868 |
| MLP | 71.231 | 0.909 | 73.539 | 1.012 | 81.115 | 0.827 | 83.050 | 0.827 |
| Norm | 11.340 | 1.014 | 13.562 | 0.886 | 14.683 | 1.140 | 15.614 | 1.178 |
| Captured (Block) | 128.503 | - | 141.480 | - | 157.038 | - | 165.331 | - |
| Block | 130.671 | 2.226 | 150.432 | 2.536 | 157.551 | 1.014 | 167.592 | 1.002 |
| % Capture (Block) | 98.341 | - | 94.049 | - | 99.674 | - | 98.651 | - |
| Embedding Layer | 1.070 | 0.420 | 1.177 | 0.439 | 1.090 | 0.412 | 1.109 | 0.424 |
| LM Head | 307.571 | 0.948 | 314.289 | 0.958 | 322.029 | 0.685 | 328.267 | 1.114 |
| Final Layer Norm | 5.327 | 0.471 | 6.259 | 0.435 | 6.993 | 0.510 | 6.993 | 0.510 |
| Captured (Model) | 3972.744 | - | 4533.818 | - | 4741.546 | - | 5028.947 | - |
| Model | 4295.739 | 80.010 | 4966.015 | 94.550 | 5156.262 | 8.688 | 5148.606 | 11.498 |
| % Capture (Model) | 92.481 | - | 91.297 | - | 91.957 | - | 97.676 | - |

Table 8: Energy of Decoder Model Components using CLEAR on RTX 6000 GPU for models(Qwen2.5-3B, Llama-3.2-3B, Gemma-3-4B, Phi3-4B) with fp16 across token length.

| Components | Qwen2.5 | | Llama3.2 | | Gemma 3 | | Phi-3 4B | |
|---|---|---|---|---|---|---|---|---|
| | Avg. | Std. Dev. | Avg. | Std. Dev. | Avg. | Std. Dev. | Avg. | Std. Dev. |
| Attention Block | 32.791 | 1.090 | 42.842 | 0.427 | 45.215 | 0.632 | 62.817 | 0.831 |
| MLP | 62.257 | 1.130 | 64.830 | 1.774 | 66.196 | 1.642 | 132.590 | 2.548 |
| Norm. (All) | 9.769 | 0.812 | 9.801 | 0.845 | 23.643 | 1.117 | 20.437 | 1.940 |
| Captured (Block) | 104.818 | - | 117.473 | - | 135.054 | - | 215.843 | - |
| Block | 113.095 | 1.855 | 121.860 | 2.067 | 149.414 | 3.911 | 214.413 | 1.106 |
| %Capture(Block) | 92.682 | - | 96.400 | - | 90.389 | - | 100.667 | - |
| Final Layer norm | 4.717 | 0.386 | 4.640 | 0.392 | 5.712 | 0.259 | 10.066 | 0.886 |
| Embedding Layer | 0.652 | 0.246 | 0.699 | 0.241 | 1.526 | 0.294 | 1.733 | 0.605 |
| LM Head | 238.235 | 0.911 | 301.543 | 1.161 | 514.961 | 1.677 | 909.761 | 3.865 |
| Captured Model | 4315.021 | - | 3718.967 | - | 5602.269 | - | 9069.265 | - |
| Model | 4489.638 | 24.041 | 3996.546 | 81.766 | 5941.196 | 72.389 | 8945.062 | 37.252 |
| %Capture (Model) | 96.111 | - | 93.055 | - | 94.295 | - | 101.389 | - |

Table 9: Energy of Decoder Model Components using CLEAR on RTX 6000 GPU for models(Qwen2.5-3B, Llama-3.2-3B, Gemma-3-4B, Phi3-4B) with fp16 and 8 token length.

| Components | Qwen2.5 | | Llama | | Gemma | | Phi | |
|---|---|---|---|---|---|---|---|---|
| | Avg. | Std. Dev. | Avg. | Std. Dev. | Avg. | Std. Dev. | Avg. | Std. Dev. |
| Attention Block | 49.584 | 0.891 | 90.613 | 0.930 | 83.220 | 1.302 | 88.889 | 0.893 |
| MLP | 132.007 | 1.069 | 149.203 | 1.259 | 144.130 | 0.776 | 143.361 | 0.984 |
| Norm. (All) | 7.068 | 0.206 | 7.310 | 0.228 | 16.947 | 1.480 | 7.736 | 0.220 |
| Captured (Block) | 188.659 | - | 247.125 | - | 244.297 | - | 239.986 | - |
| Block | 187.446 | 1.181 | 241.463 | 0.959 | 257.972 | 6.467 | 249.932 | 6.578 |
| %Capture (Block) | 100.647 | - | 102.345 | - | 94.699 | - | 96.020 | - |
| Final Layer norm | 3.358 | 0.013 | 3.454 | 0.071 | 4.021 | 0.323 | 3.604 | 0.076 |
| Embedding Layer | 0.672 | 0.256 | 0.849 | 0.034 | 1.774 | 0.046 | 1.259 | 0.549 |
| CLS + LM Head | 493.335 | 0.977 | 641.738 | 2.972 | 1112.893 | 13.443 | 1159.335 | 14.785 |
| Captured Model | 7245.422 | - | 7407.010 | - | 9889.742 | - | 9162.032 | - |
| Model | 7538.858 | 4.374 | 7724.147 | 20.110 | 10685.492 | 28.237 | 9605.711 | 30.790 |
| %Capture (Model) | 96.108 | - | 95.894 | - | 92.553 | - | 95.381 | - |

Table 10: Energy of Decoder Model Components using CLEAR on RTX 6000 GPU for models(Qwen2.5-3B, Llama-3.2-3B, Gemma-3-4B, Phi3-4B) with fp32 and 8 token length.

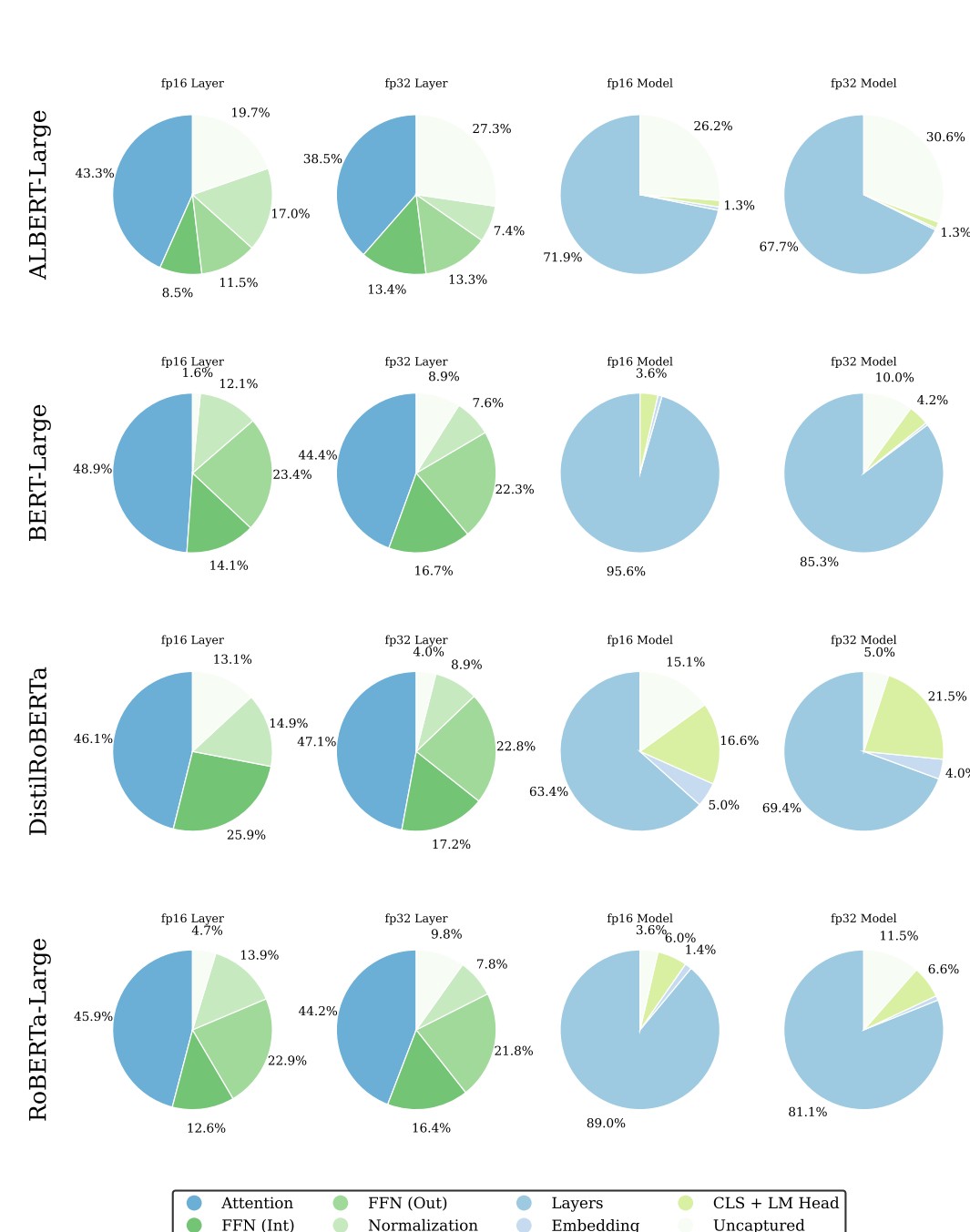

Figure 7: Energy breakdown in large variants of encoder-only models. The first two charts illustrate block-level distributions, while the latter two present distributions across the entire model. FP-16 and FP-32 precisions for each model are shown.

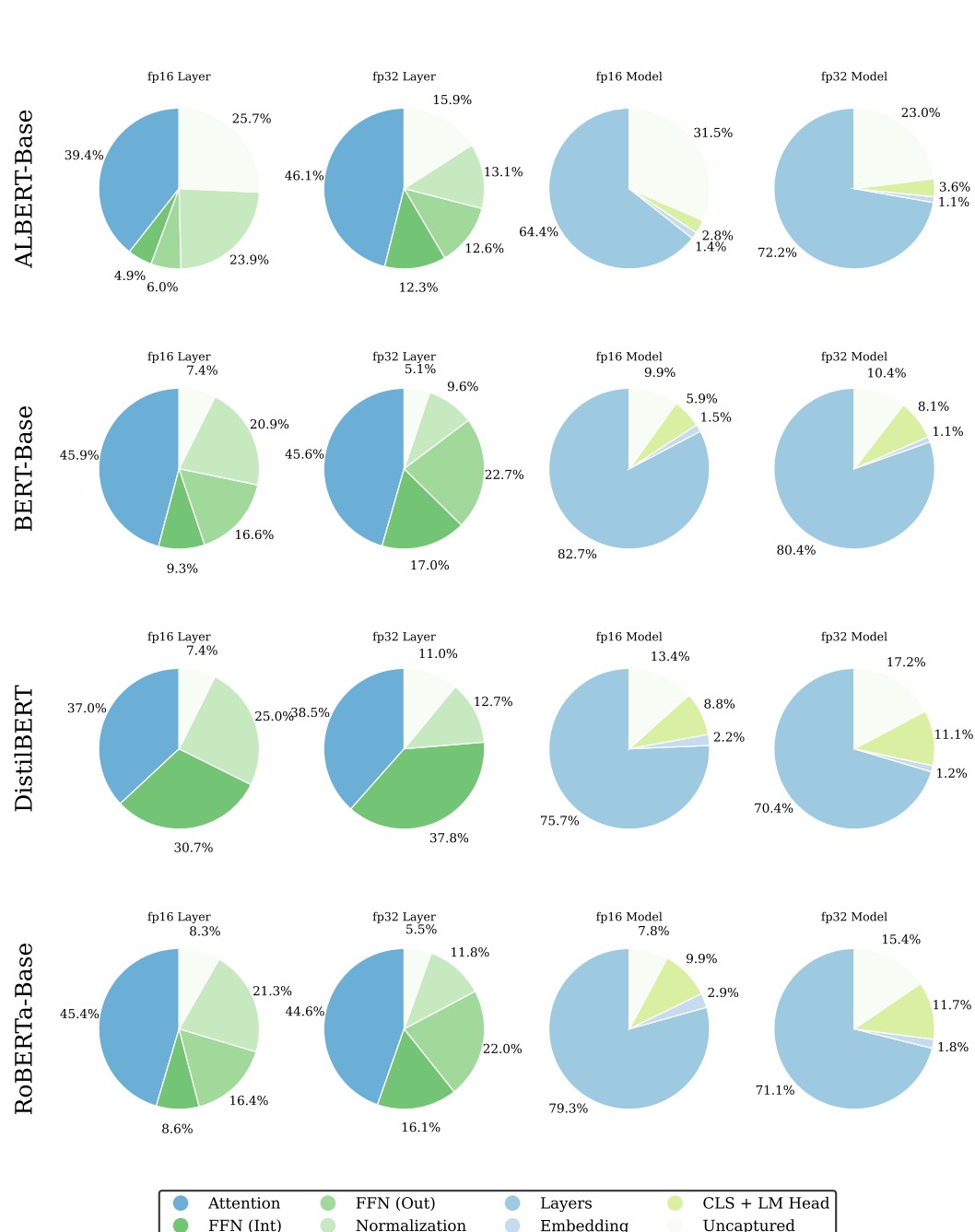

Figure 8: Energy breakdown in encoder-only models. The first two charts illustrate block-level distributions, while the latter two present distributions across the entire model. FP-16 and FP-32 precisions for each model are shown.

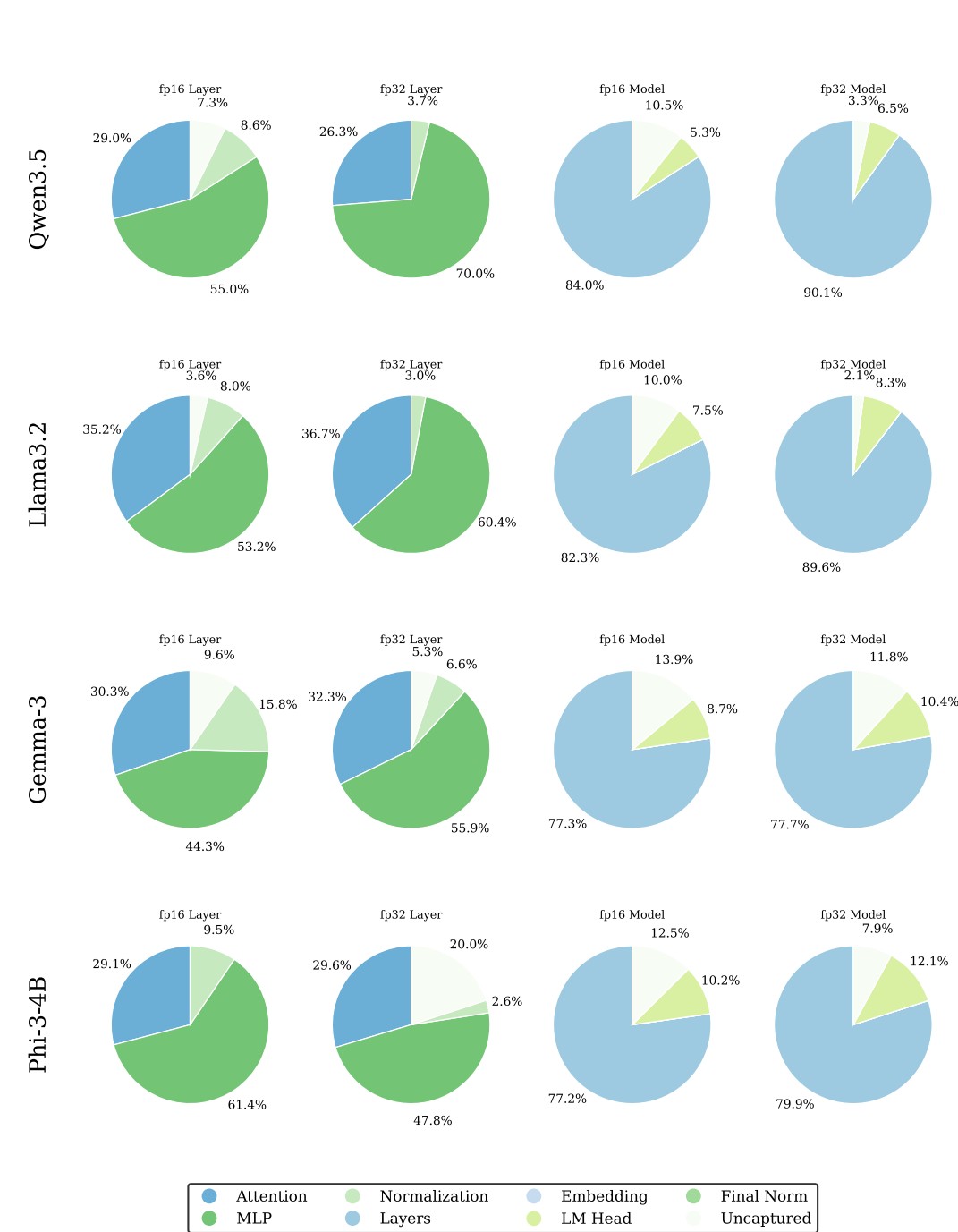

Figure 9: Energy breakdown in decoder-only models. The first two charts illustrate block-level distributions, while the latter two present distributions across the entire model. FP-16 and FP-32 precisions for each model are shown.

| Components | ALBERT | | BERT | | DistilBERT | | RoBERTa | |
|---|---|---|---|---|---|---|---|---|
| | Avg. | Std. Dev | Avg. | Std. Dev | Avg. | Std. Dev | Avg. | Std. Dev |
| Attention Block | 12.074 | 0.280 | 13.676 | 0.370 | 10.550 | 0.369 | 13.534 | 0.319 |
| FFN (Intermediate) | 1.510 | 0.441 | 2.765 | 0.192 | 8.764 | 0.442 | 2.565 | 0.321 |
| FFN (Output) | 1.848 | 0.050 | 4.944 | 0.343 | | | 4.875 | 0.401 |
| Norm. (All) | 7.317 | 0.531 | 6.216 | 0.845 | 7.135 | 0.369 | 6.352 | 0.623 |
| Captured (Block) | 22.748 | - | 27.601 | - | 26.449 | - | 27.326 | - |
| Measured (Block) | 30.627 | 0.562 | 29.802 | 0.607 | 28.551 | 0.448 | 29.797 | 0.642 |
| %Capture (Block) | 74.276 | - | 92.616 | - | 92.637 | - | 91.707 | - |
| # of Layers | 12 | - | 12 | - | 6 | - | 12 | - |
| Embedding Layer | 5.863 | 0.328 | 5.918 | 0.307 | 4.619 | 0.161 | 11.911 | 0.159 |
| CLS + LM Head | 11.687 | 0.455 | 23.666 | 0.979 | 18.396 | 1.009 | 41.081 | 0.907 |
| Captured (Model) | 385.068 | - | 387.205 | - | 194.320 | - | 410.555 | - |
| Total | 424.039 | 6.548 | 400.449 | 4.805 | 209.734 | 5.053 | 413.318 | 5.943 |
| %Capture (Model) | 90.810 | - | 96.693 | - | 92.651 | - | 99.332 | - |

Table 11: Comparison of FP16 Performance across ALBERT, BERT, DistilBERT, RoBERTa

| Components | ALBERT - Large | | BERT - Large | | Distil RoBERTa | | RoBERTa - Large | |
|---|---|---|---|---|---|---|---|---|
| | Avg. | Std. Dev | Avg. | Std. Dev | Avg. | Std. Dev | Avg. | Std. Dev |
| Attention Block | 14.861 | 0.423 | 16.903 | 0.397 | 13.226 | 0.349 | 15.547 | 0.370 |
| FFN (Intermediate) | 2.928 | 0.743 | 4.877 | 0.456 | 7.413 | 0.396 | 4.249 | 0.115 |
| FFN (Output) | 3.939 | 0.710 | 8.093 | 0.462 | | | 7.741 | 0.443 |
| Norm. (All) | 5.814 | 0.652 | 4.171 | 0.872 | 4.274 | 0.711 | 4.721 | 0.671 |
| Captured (Block) | 27.542 | - | 34.044 | - | 24.913 | - | 32.259 | - |
| Block | 34.293 | 0.493 | 34.590 | 0.709 | 28.670 | 0.511 | 33.854 | 0.833 |
| %Capture (Block) | 80.314 | - | 98.419 | - | 86.896 | - | 95.287 | - |
| # of Layers | 24 | - | 24 | - | 6 | - | 24 | - |
| Embedding Layer | 6.132 | 0.341 | 7.035 | 0.405 | 11.699 | 0.454 | 12.055 | 0.337 |
| CLS + LM Head | 12.142 | 0.671 | 30.928 | 0.831 | 39.096 | 0.908 | 52.381 | 0.721 |
| Captured (Model) | 841.313 | - | 868.134 | - | 222.813 | - | 876.937 | - |
| Total | 919.899 | 16.661 | 855.146 | 17.813 | 235.813 | 3.939 | 870.188 | 4.681 |
| %Capture (Model) | 91.457 | - | 101.519 | - | 94.487 | - | 100.776 | - |

Table 12: Comparison of FP16 Performance across variants- ALBERT Large, BERT Large, DistilRoBERTa, RoBERTa Large

| Components | ALBERT | | BERT | | DistilBERT | | RoBERTa | |
|---|---|---|---|---|---|---|---|---|
| | Avg. | Std. Dev | Avg. | Std. Dev | Avg. | Std. Dev | Avg. | Std. Dev |
| Attention Block | 18.573 | 0.517 | 19.155 | 0.436 | 17.772 | 0.412 | 17.870 | 0.327 |
| FFN (Intermediate) | 4.969 | 0.980 | 7.163 | 0.880 | 17.486 | 0.774 | 6.446 | 0.724 |
| FFN (Output) | 5.066 | 0.935 | 9.528 | 0.646 | | | 8.829 | 0.620 |
| Norm. (All) | 5.267 | 0.432 | 4.023 | 0.972 | 5.872 | 0.861 | 4.721 | 0.671 |
| Captured (Block) | 33.875 | - | 39.869 | - | 41.130 | - | 37.867 | - |
| Block | 40.282 | 0.779 | 42.030 | 0.881 | 46.199 | 0.755 | 40.054 | 1.545 |
| %Capture (Block) | 84.095 | - | 94.858 | - | 89.028 | - | 94.540 | - |
| # of Layers | 12 | - | 12 | - | 6 | - | 12 | - |
| Embedding Layer | 6.326 | 0.362 | 6.395 | 0.304 | 4.317 | 0.398 | 11.463 | 0.213 |
| CLS + LM Head | 20.468 | 0.455 | 48.188 | 0.979 | 39.096 | 0.908 | 74.482 | 1.008 |
| Captured (Model) | 510.181 | - | 558.945 | - | 320.608 | - | 566.588 | - |
| Total | 562.878 | 9.828 | 595.044 | 9.741 | 350.642 | 6.380 | 638.949 | 13.632 |
| %Capture (Model) | 90.638 | - | 93.933 | - | 91.435 | - | 88.675 | - |

Table 13: Comparison of FP32 Performance across ALBERT, BERT, DistilBERT, RoBERTa

| Components | ALBERT - Large | | BERT - Large | | Distil RoBERTa | | RoBERTa - Large | |
|---|---|---|---|---|---|---|---|---|
| | Avg. | Std. Dev | Avg. | Std. Dev | Avg. | Std. Dev | Avg. | Std. Dev |
| Attention Block | 23.380 | 0.545 | 25.777 | 0.546 | 19.888 | 0.493 | 24.569 | 0.572 |
| FFN (Intermediate) | 8.139 | 1.049 | 9.679 | 0.929 | 7.275 | 0.924 | 9.145 | 0.949 |
| FFN (Output) | 8.090 | 0.843 | 12.952 | 0.799 | 9.636 | 0.679 | 12.124 | 0.817 |
| Norm. (All) | 4.479 | 0.512 | 4.413 | 0.872 | 3.761 | 0.843 | 4.310 | 0.871 |
| Captured (Block) | 44.089 | - | 52.821 | - | 40.561 | - | 50.148 | - |
| Block | 60.662 | 0.918 | 58.006 | 0.841 | 42.234 | 1.687 | 55.608 | 1.023 |
| %Capture (Block) | 72.680 | - | 91.061 | - | 96.038 | - | 90.183 | - |
| # of Layers | 24 | - | 24.000 | - | 6.000 | - | 24.000 | - |
| Embedding Layer | 5.910 | 0.345 | 7.080 | 0.401 | 14.044 | 0.541 | 13.559 | 0.363 |
| CLS + LM Head | 20.003 | 0.500 | 62.534 | 1.021 | 75.561 | 0.936 | 97.300 | 0.746 |
| Captured (Model) | 1481.807 | - | 1461.768 | - | 343.009 | - | 1445.440 | - |
| Total | 1562.929 | 9.036 | 1485.891 | 13.015 | 350.642 | 6.380 | 1484.479 | 11.134 |
| %Capture (Model) | 94.810 | - | 98.376 | - | 97.823 | - | 97.370 | - |

Table 14: Comparison of FP32 Performance across large variants of ALBERT, BERT, DistilBERT, RoBERTa

| Component | 32 Tokens | | 64 Tokens | | 96 Tokens | | 128 Tokens | |
|---|---|---|---|---|---|---|---|---|
| | Avg. | Std. Dev. | Avg. | Std. Dev. | Avg. | Std. Dev. | Avg. | Std. Dev. |
| ALBERT-Base | | | | | | | | |
| Attention Block | 15.094 | 0.404 | 15.930 | 0.354 | 18.582 | 0.319 | 19.861 | 0.468 |
| FFN (Intermediate) | 2.306 | 0.448 | 2.205 | 0.475 | 3.341 | 0.308 | 3.650 | 0.594 |
| FFN (Output) | 2.361 | 0.423 | 2.891 | 0.599 | 4.608 | 0.837 | 4.162 | 0.751 |
| Block | 35.079 | 0.368 | 37.240 | 0.231 | 44.233 | 0.837 | 45.274 | 0.835 |
| Embedding Layer | 6.810 | 0.354 | 6.883 | 0.321 | 7.767 | 0.389 | 6.981 | 0.337 |
| CLS + LM Head | 13.384 | 0.691 | 14.438 | 0.564 | 17.171 | 0.588 | 19.246 | 0.888 |
| Captured (Model) | 441.146 | - | 468.198 | - | 555.734 | - | 569.510 | - |
| Model | 482.402 | 9.206 | 505.909 | 4.497 | 571.390 | 2.964 | 598.701 | 5.728 |
| % Capture (Model) | 91.448 | - | 92.546 | - | 97.260 | - | 95.124 | - |
| BERT- Base | | | | | | | | |
| Attention Block | 15.337 | 0.398 | 17.988 | 0.466 | 18.949 | 0.237 | 19.518 | 0.358 |
| FFN (Intermediate) | 3.472 | 0.393 | 3.695 | 0.186 | 4.628 | 0.494 | 5.626 | 0.653 |
| FFN (Output) | 5.210 | 0.374 | 5.947 | 0.361 | 7.450 | 0.290 | 8.774 | 0.272 |
| Block | 32.585 | 0.487 | 34.980 | 0.871 | 39.130 | 0.938 | 41.482 | 1.039 |
| Embedding Layer | 6.938 | 0.304 | 8.004 | 0.401 | 7.812 | 0.378 | 7.553 | 0.046 |
| CLS + LM Head | 24.876 | 0.906 | 31.174 | 0.798 | 31.497 | 0.615 | 39.772 | 0.897 |
| Captured (Model) | 422.840 | - | 458.936 | - | 508.874 | - | 545.111 | - |
| Model | 450.756 | 4.650 | 465.372 | 2.665 | 523.566 | 8.469 | 557.446 | 6.968 |
| % Capture (Model) | 93.807 | - | 98.617 | - | 97.194 | - | 97.787 | - |
| DistilBERT | | | | | | | | |
| Attention Block | 12.501 | 0.480 | 13.708 | 0.399 | 14.593 | 0.437 | 14.524 | 0.402 |
| FFN | 10.489 | 0.414 | 10.365 | 0.298 | 13.227 | 0.649 | 14.665 | 0.536 |
| Block | 32.794 | 0.791 | 34.432 | 0.775 | 38.200 | 1.049 | 39.894 | 0.579 |
| Embedding Layer | 5.322 | 0.258 | 5.503 | 0.352 | 5.583 | 0.276 | 5.043 | 0.137 |
| CLS + LM Head | 20.512 | 0.486 | 27.341 | 0.877 | 26.021 | 0.798 | 34.210 | 0.841 |
| Captured (Model) | 222.597 | - | 239.438 | - | 260.806 | - | 278.618 | - |
| Model | 234.498 | 4.466 | 245.985 | 4.135 | 279.116 | 4.422 | 304.458 | 5.907 |
| % Capture (Model) | 94.925 | - | 97.338 | - | 93.440 | - | 91.513 | - |
| RoBERTa | | | | | | | | |
| Attention Block | 14.996 | 0.398 | 17.547 | 0.438 | 18.979 | 0.426 | 18.479 | 0.332 |
| FFN (Intermediate) | 3.368 | 0.412 | 3.418 | 0.170 | 4.657 | 0.496 | 5.582 | 0.655 |
| FFN (Output) | 5.204 | 0.349 | 5.687 | 0.396 | 7.378 | 0.196 | 8.331 | 0.526 |
| Block | 32.960 | 0.703 | 35.684 | 1.005 | 40.294 | 1.569 | 41.712 | 1.503 |
| Embedding Layer | 12.383 | 0.360 | 14.455 | 0.541 | 15.189 | 0.778 | 12.794 | 0.345 |
| CLS + LM Head | 39.649 | 0.872 | 55.586 | 0.626 | 70.320 | 0.797 | 77.164 | 1.159 |
| Captured (Model) | 447.552 | - | 498.249 | - | 545.035 | - | 590.506 | - |
| Model | 470.720 | 5.510 | 506.925 | 5.836 | 553.857 | 4.894 | 594.927 | 7.070 |
| % Capture (Model) | 95.078 | - | 98.289 | - | 98.407 | - | 99.257 | - |

Table 15: Energy consumption trends with varying input token lengths tokens) for encoder only (Base) models

| Component | 32 Tokens | | 64 Tokens | | 96 Tokens | | 128 Tokens | |
|---|---|---|---|---|---|---|---|---|
| | Avg. | Std. Dev. | Avg. | Std. Dev. | Avg. | Std. Dev. | Avg. | Std. Dev. |
| ALBERT-Large | | | | | | | | |
| Attention Block | 17.730 | 0.472 | 19.696 | 0.645 | 22.541 | 0.529 | 24.279 | 0.435 |
| FFN (Intermediate) | 3.364 | 0.812 | 3.399 | 0.927 | 5.134 | 0.888 | 4.352 | 0.821 |
| FFN (Output) | 4.117 | 0.688 | 4.724 | 0.900 | 5.424 | 0.956 | 6.064 | 0.913 |
| Block | 39.912 | 0.474 | 44.930 | 1.058 | 51.089 | 0.969 | 55.649 | 0.597 |
| Embedding Layer | 7.881 | 0.163 | 7.846 | 0.417 | 7.849 | 0.397 | 7.914 | 0.383 |
| CLS + LM Head | 14.461 | 0.671 | 15.293 | 0.715 | 17.528 | 0.609 | 22.213 | 0.457 |
| Captured (Model) | 980.23 | - | 1101.46 | - | 1251.52 | - | 1365.71 | - |
| Model | 1029.60 | 5.731 | 1151.70 | 20.540 | 1320.18 | 9.018 | 1394.97 | 7.538 |
| % Capture (Model) | 95.205 | - | 95.638 | - | 94.799 | - | 97.903 | - |
| BERT- Large | | | | | | | | |
| Attention Block | 18.641 | 0.275 | 20.986 | 0.405 | 23.092 | 0.670 | 24.792 | 0.351 |
| FFN (Intermediate) | 5.424 | 0.208 | 6.216 | 0.630 | 7.962 | 0.797 | 7.189 | 0.766 |
| FFN (Output) | 8.593 | 0.491 | 9.499 | 0.541 | 12.091 | 0.859 | 12.533 | 0.752 |
| Block | 37.557 | 0.942 | 41.864 | 1.259 | 48.511 | 1.519 | 50.453 | 1.124 |
| Embedding Layer | 7.729 | 0.311 | 8.106 | 0.413 | 8.131 | 0.403 | 8.145 | 0.326 |
| CLS + LM Head | 32.645 | 0.943 | 39.421 | 0.926 | 41.166 | 0.892 | 40.203 | 0.470 |
| Captured (Model) | 941.732 | - | 1052.26 | - | 1213.56 | - | 1259.22 | - |
| Model | 932.191 | 13.932 | 1060.72 | 28.963 | 1206.09 | 13.967 | 1276.73 | 8.691 |
| % Capture (Model) | 101.024 | - | 99.202 | - | 100.620 | - | 98.628 | - |
| Distil RoBERTa | | | | | | | | |
| Attention Block | 15.753 | 0.276 | 17.618 | 0.401 | 18.913 | 0.426 | 18.517 | 0.365 |
| FFN | 3.499 | 0.374 | 3.542 | 0.343 | 4.595 | 0.508 | 5.488 | 0.679 |
| Block | 5.287 | 0.354 | 5.664 | 0.450 | 7.623 | 0.423 | 8.228 | 0.474 |
| Embedding Layer | 33.617 | 0.759 | 35.723 | 1.081 | 39.958 | 1.678 | 41.404 | 1.463 |
| CLS + LM Head | 12.882 | 0.307 | 14.689 | 0.416 | 14.714 | 0.358 | 12.636 | 0.346 |
| Captured (Model) | 39.648 | 0.854 | 55.718 | 0.776 | 70.263 | 0.816 | 77.483 | 1.026 |
| Model | 254.234 | - | 284.747 | - | 324.726 | - | 338.543 | - |
| % Capture (Model) | 270.513 | 4.867 | 292.914 | 3.888 | 319.139 | 5.044 | 341.740 | 5.520 |
| RoBERTa-Large | | | | | | | | |
| Attention Block | 18.648 | 0.346 | 20.390 | 0.353 | 22.573 | 0.386 | 24.780 | 0.474 |
| FFN (Intermediate) | 5.437 | 0.219 | 5.567 | 0.131 | 7.809 | 0.834 | 7.066 | 0.813 |
| FFN (Output) | 8.476 | 0.553 | 9.417 | 0.539 | 11.755 | 0.691 | 12.475 | 0.750 |
| Block | 36.273 | 0.979 | 40.831 | 1.651 | 47.154 | 1.870 | 50.343 | 1.970 |
| Embedding Layer | 14.251 | 0.372 | 14.534 | 0.406 | 14.647 | 0.411 | 14.609 | 0.391 |
| CLS + LM Head | 51.022 | 0.958 | 69.201 | 1.015 | 89.032 | 1.128 | 94.427 | 0.949 |
| Captured (Model) | 935.819 | - | 1063.67 | - | 1235.36 | - | 1317.27 | - |
| Model | 965.216 | 9.228 | 1054.90 | 6.207 | 1238.24 | 10.077 | 1347.08 | 23.636 |
| % Capture (Model) | 96.954 | - | 100.831 | - | 99.768 | - | 97.787 | - |

Table 16: Energy consumption trends with varying input token lengths tokens) for encoder only (Large) models

