# OpenReview forum: "Dissecting Transformers: A 'CLEAR' Perspective towards Green AI"
_ICLR.cc/2026/Conference — Submitted to ICLR 2026_

### Official Review · Reviewer_vHyf · 2025-10-24

**Soundness:** 3
**Presentation:** 3
**Contribution:** 1
**Rating:** 4
**Confidence:** 3

**Summary:**

This paper provides an approach to tracking the environmental cost of large language models. The inference latency and flops reports coarse, model-level energy metrics, which obscure the contribution of individual architectural components. To bridge this gap, the paper introduces Component-Level Energy Assessment via Repeated sampling (CLEAR), a method to measure the inference energy consumption of fine-grained components within Transformer architectures.

**Strengths:**

1. **Practical Method**: CLEAR is a practical approach that produces an accurate component-level energy assessment.
2. **Insights**: The findings provide insights into the relationship between energy and floating-point precision, input token length, and flops.
3. **Clarity**: The paper is well-structured and clearly explains the method and conclusion. The commitment to releasing the code enhances reproducibility.

**Weaknesses:**

1. **Lack of scientific contribution**: CLEAR is a simple measuring method that averages over multiple samples, which is more like a common practice in engineering rather than scientific discovery. Although the motivation is interesting, it has little influence on the energy measuring approach or insights into efficient architecture design.
2. **Limited Insights on Model Architecture Design**: While the paper identifies the energy cost of each component in the Transformer architecture, it lacks an analysis of how to design a more efficient architecture. A more in-depth discussion or experiment would further strengthen the work.
3. **Limited Hardware**: The study is conducted exclusively on NVIDIA Ada-Lovelace GPUs using the NVML interface. The findings and the precise behavior of CLEAR may not generalize directly to other hardware (e.g., other GPU vendors, TPUs, or Arm devices). These devices may have different power management, sensor granularity, and architectural optimizations.

**Questions:**

1. **Dependency on Implementation**: Is the energy cost related to the implementation, such as Flash Attention, fused CUDA kernel, PyTorch compile, etc?

---

> ### Author Response · Authors · 2025-11-20
>
> We sincerely thank the reviewer for their thoughtful feedback and for recognizing the practicality, accuracy, and clarity of our methodology and its valuable insights. We have provided clarifications addressing the identified weaknesses.
>
> &nbsp;
>
> ### **W1. Clarification on CLEAR's Contributions**
>
> In our work, we propose CLEAR, a simple and easily adoptable Methodology that dissects the Transformer Architecture to reliably measure the energy consumption of individual components and amplifies the energy signal provided by the underlying GPU energy sensor. The amplification of the energy signal is achieved by repetitions to minimize the excessive idle energy that may be calculated given the coarse grained energy sensor. Simple components such as Normalization and Attention block take < 1 ms to complete the execution whereas sampling time frame of the sensor is relatively large (about 20-50 ms). A number of previous works rely heavily on two methodologies,
>
> 1) Methodologies that estimate energy at a coarse granularity often by measuring long sequences over large datasets and averaging to obtain per-token or fine-grained approximations [1][2][3]. For example, Attentions Under the Microscope [3] reports the energy values over large datasets at the scale of MJ which fails to reliably isolate the true fine-grained effect at Attention level.
>
> 2) Hardware-centric approaches that rely on execution tracing tightly coupled with hardware-specific additional sensors or system instrumentation [8][9][5][4]. In such a methodology, access to physical GPU hardware becomes necessary and the additional sensor used needs to have low latency and high precision to reliably track/trace the function execution. Along with that, hardware based approaches are not scalable as they depend on specialized equipment and physical access to every target device making them difficult to deploy widely across large systems with diverse hardware environments.
>
>
> Our methodology, though simple, effectively overcomes the limitations of existing energy-measurement approaches. CLEAR operates at a software/code-module level rather than at a hardware level that allows our methodology to be independent of the underlying GPU and the sensor granularity. Along with amplification of the energy signal to overcome the time frame of the sensor, the amplification strategy serves the purpose of reducing the error induced by idle energy as well. As mentioned in Section 3.2, though the number of repetitions used remains large, the error due to idle energy (about 70mJ per second) will be implicitly measured for a maximum of about 20-50 ms. By normalizing energy measurement for N repetitions, the error induced by the idle energy reduces as N → (a large number) Infinity increasing the precision of our Methodology.
>
> CLEAR is an easily adoptable, requires no hardware modifications, and is both hardware- and model-agnostic, addressing a major limitation acknowledged in the literature [5][7][8]. We will incorporate these clarifications in the revised manuscript.
>
> ---
> [1] IrEne: Interpretable Energy Prediction for Transformers (https://arxiv.org/abs/2106.01199)
>
> [2] Green AI (https://arxiv.org/abs/1907.10597 )
>
> [3] Attentions Under the Microscope: A Comparative Study of Resource Utilization for Variants of Self-Attention (https://arxiv.org/abs/2507.07247 )
>
> [4] How Hungry is AI? Benchmarking Energy, Water, and Carbon Footprint of LLM Inference (https://arxiv.org/abs/2505.09598 )
>
> [5] PowerSensor3: A Fast and Accurate Open Source Power Measurement Tool (https://arxiv.org/abs/2504.17883 )
>
> [6] CPPJoules: An Energy Measurement Tool for C++ (https://arxiv.org/abs/2412.13555)
>
> [7] Part-time Power Measurements: nvidia-smi’s Lack of Attention (https://arxiv.org/abs/2312.02741 )
>
> [8] Understanding GPU Power: A Survey of Profiling, Modeling, and Simulation Methods (https://dl.acm.org/doi/10.1145/2962131 )
>
> [9] MLEE: Method Level Energy Estimation — A machine learning approach (https://www.sciencedirect.com/science/article/pii/S2210537921000822 )

---

> > ### Comment · Reviewer_vHyf · 2025-11-27
> > **Questions on the contribution**
> >
> > I agree that CLEAR is a simple yet effective method to measure the comsumption in the transformer architecture. However, I believe the proposed approach is a common approach in engineering, including averaging, per layer profiling, etc. For example, a professional AI company needs to measure the energy cost precisely in order to pricing the API service, and averaging the energy cost on multiple inference is the most intuitive way to do so. In fact, the author(s) could contribute to this area from another perspective, e.g., concluding which hyperparameters are the best choice and analyzing the reason, or providing a helpful and user-friendly python packages to automatically measure precise energy cost. The method itself is so simple that the paper lacks a valuable takeaway, and other Reviewers have the same feeling. I believe this work is a good start, but not qualified as a conference paper right now.

---

> ### Author Response · Authors · 2025-11-20
>
> ### **W2. Insights on Model Architecture Design**
>
> The central contribution of our work is to provide a reliable methodology for measuring the energy consumption and to empirically observe the non-trivial relation between FLOPs and Energy Consumption for different components of Transformer architecture. Through this work, we establish the missing prerequisite required for principled, energy-aware design of transformer architectures. Our contributions not only fill a critical methodological gap but also provides a strong motivation for the following two paradigms:
>
> 1) With fine-grained energy measurement, model-architecture designers can directly quantify the energy usage of individual components, enabling implementation-level optimizations that can be integrated back into the model. CLEAR allows designers to validate architectural tradeoffs and test energy-aware redesign strategies. Additionally, we have experimented with different Attention Variants and Optimizations and results have been updated in our manuscript in **Appendix D**. Similar studies can motivate design of energy efficient Transformer components.
>
>
> 2) CLEAR provides the formal foundation for energy prediction modelling. CLEAR enables to record reliable and accurate ground-truth values that can be used for predictive modelling. While complete architecture design is a multi-stage pipeline requiring (i) reliable ground truth, (ii) scaling laws, and (iii) design-space exploration, our work addresses the first step and provides empirical insights for Scaling Laws and optimization impacts. To build upon works like IrEne [1], CLEAR can enhance their energy prediction approach by providing reliable and accurate ground truths for larger scale of models, and provide empirical insights on impact of input and output token scaling on Energy for better predictive modelling.
>
> Although how to design a more efficient architecture lies outside the present scope of our work, CLEAR provides a strong empirical foundation required for it.
>
> &nbsp;
>
> ### **W3.Clarification on CLEAR's Hardware Agnostic Nature**
>
>
> While we acknowledge the limitation that our experiments were performed on a single hardware device type (an NVIDIA Ada-Lovelace GPU), our methodology is portable and operates entirely at a software layer. As a result, CLEAR is agnostic to the underlying hardware and the granularity of built-in energy sensors, and it can be readily extended to other hardwares such as TPUs, AMD GPUs, or Arm-based devices. Due to limited hardware availability, we weren’t able to validate CLEAR’s approach on additional hardware devices. However, as a future work, conducting empirical analysis of different architectural components across diverse hardware devices using CLEAR could yield potentially interesting insights.
>
> &nbsp;
>
> ### **Q1. Additional Findings on Attention Variants and Optimizations**
>
> We run additional experiments to check the energy consumed by different variants and optimizations of the Attention Mechanism and have added a detailed discussion in Appendix D of our updated manuscript. Specifically we experiment with three Attention implementations, Flash Attention, Eager Attention and SDPA and observe that energy consumption follows the trend of SDPA ~ Flash Attn. < Eager Attn. Additionally, we experiment with different input sequence lengths and observe that energy consumption does not increase in direct proportion to FLOPs. Instead, the measurements fit the relationship given by the equation $E = E_0 + k * FLOPs$, where $E_0$ includes fixed costs of kernel launch overhead, memory allocation, and warm up and compilation steps.
>
> Similarly optimizations such as Torch Compile, Max Autotune and Reduce Overhead are able to reduce the energy consumption by approximately 30%, 55% and 55% respectively for the Qwen2.5-3B model and about 48%, 64% and 64% respectively for the Gemma3-4B model. Refer  **Appendix D** for detailed insights.
>
> &nbsp;
>
> ---
> [1] IrEne: Interpretable Energy Prediction for Transformers (https://arxiv.org/abs/2106.01199)
>
> ---
>
> &nbsp;
>
> Thank you again for your thoughtful comments and suggestions. We hope we’ve addressed all your concerns. We are happy to clarify any additional questions you have and engage in a follow-up discussion.

---

### Official Review · Reviewer_uuyn · 2025-10-29

**Soundness:** 4
**Presentation:** 3
**Contribution:** 4
**Rating:** 8
**Confidence:** 5

**Summary:**

This paper presents a fine-grained empirical analysis of inference energy across different components of transformer architecture, and proposes a CLEAR approach that aims to bridge the temporal mismatch between component execution and energy measurements.

They test 15 models spanning and show that they approach captures more than 90% of the model’s total energy as individual components.

Their analysis shows that Attention blocks use more energy per FLOP, and that FLOPs alone don't capture the true energy cost at a
a component level, which is important because it's an approach that is often used by the community as a proxy for energy estimation.

**Strengths:**

This paper tackles an important challenge that arises from the temporal mismatch between the speed of component execution for Transformers and the energy monitoring approaches that are used to measure it.

They look at 4 different classes of Tranformer-based models, not only Decoder-only models that currently most popular, and including MoX variants, which are increasingly used.

The way in which their methodology is described and results presented is clear and easy to read, and the code is readily available for validation and reproducibility.

The results are compelling and the experiments are done in a methodologically-sound way.

**Weaknesses:**

-Some related work is overlooked, e.g https://aclanthology.org/2021.acl-long.167.pdf and https://github.com/ml-energy/zeus

- Unless I missed something, there isn't a full list of models tested in one place? I saw the figures in the Appendix, but I would appreciate an overview table with the model names and parameter counts.

- While the approach is very interesting and valid, what's missing for me is the next steps or takeaways - can CLEAR be used to create new types of Transformer-based architectures that are more efficient? can it help developers identify energy sinks or hotspots?

**Questions:**

- Why not test models for a set time (e.g. 30 seconds) instead of a a set number of repetitions?

- What kind of actionable insights can your work yield for AI practitioners? i.e. how does it help someone to know that a certain component of a Transformer model consumes more/less energy? "Attention is the most computationally expensive sub-component" is difficult to put into practice since it's the main building block of Transformer models to begin with..

- Why not test different sizes of the same model to validate your findings more in depth? e.g. different versions of LLaMa or Qwen?

---

> ### Author Response · Authors · 2025-12-03
>
> We sincerely thank the reviewer for their exceptionally positive evaluation of our work and for recognizing the soundness of our methodology, the compelling nature of our results, and the importance of the problem we address.
>
> &nbsp;
>
> ### **W1. Model Overview table**
>
> We appreciate reviewer's suggestion and will add a comprehensive table listing every model tested along with parameter counts and key configuration details in the camera ready version of the manuscipt.
>
> &nbsp;
>
> ### **W2. Implications**
>
> We agree that the ultimate goal of energy measurement is to inform better design. While proposing new Transformer architectures lies beyond the scope of our current study, we believe CLEAR establishes a necessary prerequisite for principled, energy-aware model design. By reliably isolating and quantifying energy consumption at the component level, CLEAR provides ground-truth measurements that enable researchers and practitioners to evaluate architectural trade-offs with precise measurements rather than relying on coarser approximations.
>
> CLEAR provides a strong formal foundation for predictive modelling for energy consumption. CLEAR enables to record reliable and accurate ground-truth values that can be used for predictive modelling. While complete architecture design is a multi-stage pipeline requiring (i) reliable ground truth, (ii) scaling laws, and (iii) design-space exploration, our work addresses the first step and provides empirical insights for Scaling Laws and optimization impacts. To build on works like IrEne [1], CLEAR can enhance their energy prediction approach by providing reliable and accurate ground truths at larger scale of implementation, and provide insights on impact  of input and output token scaling on Energy for better predictive model.
>
> We performed additional experiments to benchmark the energy consumed by different variants and optimizations of the Attention Mechanism and have added a detailed discussion in Appendix D of our updated manuscript. Specifically we experiment with three Attention implementations, Flash Attention, Eager Attention and SDPA and observe that energy consumption follows the trend of SDPA ~ Flash Attn. < Eager Attn. Additionally, we experiment with different input sequence lengths and observe that energy consumption does not increase in direct proportion to FLOPs. Instead, the measurements fit the relationship given by the equation $ E = E_0 + k * FLOPs $, where $ E_0 $ accounts for fixed energy costs of kernel launch overhead, memory allocation, and warm up and compilation steps.
>
> Similarly optimizations such as Torch Compile, Max Autotune and Reduce Overhead are able to reduce the energy consumption by approximately 30%, 55% and 55% respectively for the Qwen2.5-3B model and about 48%, 64% and 64% respectively for the Gemma3-4B model. Refer to Appendix D for more insights.
>
> &nbsp;
>
> [1] IrEne: Interpretable Energy Prediction for Transformers (https://arxiv.org/abs/2106.01199)

---

> ### Author Response · Authors · 2025-12-03
>
> ### **Q1. Fixed Time vs. Fixed Repetitions**
>
> We believe that running  CLEAR for a fixed amount of time instead of fixed number of repetitions would essentially yield the same results. In CLEAR, we selected a fixed number of repetitions (10,000) specifically to ensure the total experiment duration (e.g., 13.86 seconds for Gemma's MLP layer with 10,000 repetitions) is orders of magnitude larger than the NVML’s 20-50 ms sampling rate. This long duration is enough to ensure a stable and reliable average energy per repetition.
>
> &nbsp;
>
> ### **Q2. Insights**
>
> The central contribution of our work is to provide a reliable methodology for measuring the energy consumption and to empirically observe the non-trivial relation between FLOPs and Energy Consumption for different components of Transformer architecture. Through this work, we establish the missing prerequisite required for principled, energy-aware design of transformer architectures. Our contributions not only fill a critical methodological gap but also provides a strong motivation for the following two paradigms:
>
> 1. With fine-grained energy measurement, model-architecture designers can directly quantify the energy usage of individual components, enabling implementation-level optimizations that can be integrated back into the model. CLEAR allows designers to validate architectural tradeoffs and test energy-aware redesign strategies. Additionally, we have experimented with different Attention Variants and Optimizations and results have been updated in our manuscript in Appendix D. Similar studies can motivate design of energy efficient Transformer components.
>
> &nbsp;
> 2. CLEAR provides the formal foundation for energy prediction modelling. CLEAR enables to record reliable and accurate ground-truth values that can be used for predictive modelling. While complete architecture design is a multi-stage pipeline requiring (i) reliable ground truth, (ii) scaling laws, and (iii) design-space exploration, our work addresses the first step and provides empirical insights for Scaling Laws and optimization impacts.
> Although how to design a more efficient architecture lies outside the present scope of our work, CLEAR provides a strong empirical foundation required for it.
>
> Although Attention is central to Transformer architectures, CLEAR’s purpose is not to replace it but to identify it as an energy hotspot and allow future optimizations by reliable energy measurements during model architecture design. Instead of the generic claim that “Attention is expensive,” CLEAR reveals how different variants, sequence lengths, KV-cache and runtime optimization settings yield lesser energy consumption, enabling targeted, energy-aware improvements without altering the core architecture.
>
> &nbsp;
>
> ### **Q3.  Model Size Variants**
>
> We currently perform an empirical analysis across 15 models from 4 different architecture families. Specifically for Encoder-only models, we conduct analysis on BERT / BERT-Large , RoBERTa / RoBERTa-Large and ALBERT/ ALBERT-Large pairs to study the impact of model size. We are conducting experiments on 3 variants of the Qwen 2.5 family (Decoder-based Models) (specifically for 1.5B, 3B, and 7B model sizes) and will include the additional results, along with an analysis of how our findings scale with model size, in the revised version of the draft.
>
> &nbsp;

---

### Official Review · Reviewer_AfSZ · 2025-11-01

**Soundness:** 3
**Presentation:** 3
**Contribution:** 2
**Rating:** 4
**Confidence:** 3

**Summary:**

This paper studies the energy consumption of large language models (LLMs) at an individual component (operator) level, such as attention, MLP, layer norm, embedding layer, LM head, etc. In order to measure the energy consumption of small components, the paper proposes to repeat measurements many times and calls this methodology CLEAR (Component-Level Energy Assessment via Repeated sampling). The study found that the proposed approach can capture most of the energy consumption of a full model. It also found that the energy consumption per token varies significantly for individual components, and that the FLOP is not a good measure of energy consumption.

**Strengths:**

The energy consumption of moden LLMs is one of the most important practical concerns that need to be addressed as LLMs are widely deployed at scale. I believe that understanding the energy efficiency of individual building blocks of an LLM will be an important first step in designing more energy efficient model architectures - Knowing how individual components affect the energy consumption, AI model designers can try to use more energy efficient components. In that sense, I think the paper studies an important and timely problem.

While not surprising, the main take-away points of the paper - that the number of floating-point operations (FLOPs) is not a realiable measure of a component's energy consumption and the energy consumption varies significantly for different components will be valuable for model designers to know.

**Weaknesses:**

The proposed method to repeat (replay) a short/fast operator multiple times to more reliably measure its power consumption is somewhat obvious and has been used in many domains. For example, in order to measure the energy consumption of individual microprocessor instructions, people typically run the same instruction many times. In that sense, I do not think the repeated measurement itself can be considered as a main technical contribution. In my opinion, the paper's main contributions come from the experimental validation of this approach for an LLM and the findings from experimental results.

While the proposed methodology appears to work well enough for the specific setups used in the paper, It is not clear if simply repeating an operator multiple times will work well for production systems with optimizations such as speculative execution and caching of KV cache values. It will be helpful if the paper can provide more clear discussions and experimental results on which aspects of a component's operations are indeed repeated and where the repeated executions may differ from the exeuction when running a full model. In particular, data movement overhead (for example, KV cache movements) can be significantly different between a single execution and multiple repeated executions - for example, if data is cached, repeated executions will be faster and require less data movement.

While the main findings are interesting, it is not surprising that the energy consumptions vary significantly among different components and that FLOPs do not reliably represent energy consumption. The paper will be much stronger if it can more concretely show how its findings can be used or what their implications are for designing more efficiency AI models and systems. For example, did the experimental results found certain types of components will be better or certain types of model architecture willl be more efficient? Can the component level energy consumption results be used to predict which model architecture will be more energy efficient or inspire new building blocks? If FLOPs are not a good metric to use, is there a better way to estimate the energy consumption w/o measuring on a real system? Does the cost of individual components change for different types of hardware? I would suggest adding more concrete discussions on the impact of the experimental findings.

**Questions:**

See the questions above.

---

> ### Author Response · Authors · 2025-11-22
>
> We sincerely thank the reviewer for their feedback. We appreciate their validation that this is a timely and important study, and we are glad they found our main conclusions regarding component-level energy variation to be valuable.
>
> &nbsp;
>
> ### **W1. Clarification on Contribution**
>
> We clarify that the central contribution of CLEAR is a reliable, model agnostic methodology to precisely measure the true energy consumption of fine-grained Transformer components like Attention, MLP and LM Head and its validation across 4 architecture families and 15 models. CLEAR comprises three coordinated steps: 1) an Activation Store to cache the activations and isolate component specific execution 2) Energy Signal Amplification Strategy and 3) a Validation Approach.
>
> Although the high-level idea of Energy Signal Amplification may seem simple, its adoption to Transformer models is non-obvious. Low-level Instruction energy measurement amplifies a single, independent assembly instruction whose execution is uniform and hardware-deterministic.  In contrast, specific components of Transformer models are built as a complex composite of multiple heterogenous functions that combine memory movements, dynamic kernel dispatch, computational operations and model-specific implementational variations. These functions do vary depending on the model features (eg: hidden dimensions, up & down projections,etc), the underlying GPU, kernel libraries and runtime optimizations. Crucially, repeated execution may not trivially preserve kernel structure across repeated runs. For example, Torch Compile fuses kernels differently depending on the surrounding execution graph. Similarly, Reduced Overhead alters synchronization paths and removes profiling kernels.
>
> Additionally, architectural components in Transformer models exhibit intrinsic coupling that makes isolated replay non-trivial. For example, during multi-token output generation, models store the key and value activations in a KV Cache which causes temporal coupling between different parts of execution. Because of such complexities, it is not evident that simply repeating a component would yield a stable and faithfully amplified energy signal.
>
> Our experiments show that CLEAR consistently captures >90% of component energy across 15 diverse models, with strong cross-trial stability and predictable scaling even under temporal dependencies introduced by KV Cache. CLEAR provides a methodological pipeline that combines Energy Signal Amplification with Component isolation, Activation Store and Validation metrics.
>
> &nbsp;

---

> > ### Author Response · Authors · 2025-11-22
> >
> > ### **W2.  Clarification on effect of  KV Cache**
> >
> > We acknowledge the reviewer’s concern about the adaptability of CLEAR to the production-setting environment, in presence of optimizations like KV Cache reuse. We clarify how CLEAR addresses the concern and provide additional experiments in **Appendix Section E**.
> >
> > In CLEAR, for amplification, the component (e.g., an MLP or Attention block) is isolated and executed with the same input activation tensor repeatedly, without altering the rest of the model or invoking a full forward pass. This ensures Arithmetic operations are repeated exactly as in a real execution and same memory footprints are preserved across repetitions.
> >
> > We acknowledge that data movement overheads particularly affected by KV Caching may behave differently under amplification. To isolate the effect of KV Cache on energy consumption, we conducted additional experiments :
> >
> > 1) Using Qwen2.5-3B model (FP16 Precision), we capture the energy consumption of the entire model across the Prefill phase, Decode phase, and complete Output Generation pipeline for a single new token. We observe that the Prefill stage and Output Generation [using model.generate()] show a linear increase in energy consumption and FLOPs with increasing input sequence length. However, the energy consumed by the Decode phase remains almost invariant to input sequence length due to KV cache reuse for additional token generation. KV cache eliminates redundant computation by retrieving previously computed Keys and Values in the Attention block. (Refer to Table 7 and Figure 5 in Appendix Section E.)
> >
> > 2) We additionally run experiments to isolate the effect of KV Cache in the Attention Block of the decode stage. We execute a trial prefill phase run to store the Key and Value activations and use them to enable KV Cache optimization at the Attention Block. On comparing the energy consumption of Attention block with and without KV Cache, we empirically observe that energy increases with an increase of input token length as block performs full O(T^2) attention. However, with KV Cache enabled, the energy consumed by the Attention block remains invariant to the input sequence length. (Refer to Table 6 and Figure 5 in Appendix Section E.)
> >
> > We hope that the additional experiments offer a clearer understanding of how caching effects are isolated and influence energy consumption by CLEAR’s methodology. Additional experiments on multi-token generation and KV-cache sensitive behavior are included in Appendix Section E, and will be incorporated into the main text in future revisions.

---

> > > ### Author Response · Authors · 2025-11-22
> > >
> > > ### **W3. Implications and Impact of Experimental Findings**
> > >
> > >
> > > Based on additional experiments, we observe that the Energy Consumption of the Prefill phase does not scale in direct proportion with either the input token sequence length or the associated FLOPs. In contrast, during multi-token generation, the Decode phase exhibits nearly constant energy consumption per newly generated token, primarily due to the reuse of the KV cache. We also note a marginal increase in energy as the length of the previously generated sequence grows. This slight increase is attributable to the expanding KV tensors, which lead to higher energy-intensive memory movement operations.
> > >
> > > 1) Did the experimental results find that certain types of components are better or certain types of model architecture will be more efficient?
> > >
> > > → We run additional experiments to check the energy consumed by different variants and optimizations of the Attention Mechanism and have added a detailed discussion in Appendix D of our updated manuscript. Specifically we experiment with three Attention implementations, Flash Attention, Eager Attention and SDPA and observe that energy consumption follows the trend of SDPA ~ Flash Attn. < Eager Attn. Additionally, we experiment with different input sequence lengths and observe that energy consumption does not increase in direct proportion to FLOPs. Using CLEAR, we observe that optimizations such as Torch Compile, Max Autotune and Reduce Overhead are able to reduce the energy consumption by approximately 30%, 55% and 55% respectively for the Qwen2.5-3B model and about 48%, 64% and 64% respectively for the Gemma3-4B model (Refer to Appendix D for more insights.) Similar studies can motivate design of energy efficient Transformer components.
> > >
> > > 2) Can the component level energy consumption results be used to predict which model architecture will be more energy efficient or inspire new building blocks?
> > >
> > > → CLEAR provides a strong formal foundation for predictive modelling for energy consumption. CLEAR enables to record reliable and accurate ground-truth values that can be used for predictive modelling. While complete architecture design is a multi-stage pipeline requiring (i) reliable ground truth, (ii) scaling laws, and (iii) design-space exploration, our work addresses the first step and provides empirical insights for Scaling Laws and optimization impacts. To build on works like IrEne [1], CLEAR can enhance their energy prediction approach by providing reliable and accurate ground truths at larger scale of implementation, and provide insights on impact  of input and output token scaling on Energy for better predictive model.
> > >
> > > &nbsp;
> > > [Continued]
> > >
> > > ---
> > >
> > > [1] IrEne: Interpretable Energy Prediction for Transformers (https://arxiv.org/abs/2106.01199)

---

> > > > ### Author Response · Authors · 2025-11-22
> > > >
> > > > 3) If FLOPs are not a good metric to use, is there a better way to estimate the energy consumption w/o measuring on a real system?
> > > >
> > > > → We do not completely eliminate that FLOPs are not a good metric; rather empirically observe that FLOPs alone are insufficient to accurately estimate the energy consumption of Transformer components. Prior works ([1]) have explored metrics like model features (number of dimensions, layers, vocabulary size, etc), but for component-level variants and optimizations, extensive and reliable Ground truth energy measurements are essential for reliable energy estimation. For example, for multi-token generation, a predictive model based on FLOPs or coarse model features may predict a constant marginal energy per additional token generated. However, such an approach may overlook the subtle increase in energy with increasing number of generated tokens even though the total FLOPs consumed remains the same.
> > > >
> > > > Furthermore, existing methodologies often estimate energy at a coarse granularity typically by measuring long sequences over large datasets and averaging to obtain per-token or fine-grained approximations [1][2][3]. For example, [3] reports the energy values for large datasets at the scale of MJ which fails to reliably isolate the true fine-grained effect at Attention level.
> > > >
> > > >
> > > > 4) Does the cost of individual components change for different types of hardware?
> > > >
> > > > → While we acknowledge in the limitations section that our experiments were performed on a single hardware device type (an NVIDIA Ada-Lovelace GPU), our methodology is portable and operates entirely at a software/code module level. As a result, CLEAR is agnostic to the underlying hardware and the granularity of built-in energy sensors, and it can be readily extended to other hardwares such as TPUs, AMD GPUs, or Arm-based devices. Due to limited hardware availability, we weren’t able to validate CLEAR’s approach on additional hardware devices. However, as a future work, conducting empirical analysis of different architectural components across diverse hardware devices using CLEAR could yield potentially interesting insights.
> > > >
> > > > &nbsp;
> > > >
> > > > ---
> > > >
> > > > [1] IrEne: Interpretable Energy Prediction for Transformers (https://arxiv.org/abs/2106.01199)
> > > >
> > > > [2] Green AI (https://arxiv.org/abs/1907.10597 )
> > > >
> > > > [3] Attentions Under the Microscope: A Comparative Study of Resource Utilization for Variants of Self-Attention (https://arxiv.org/abs/2507.07247 )
> > > >
> > > > ---
> > > >
> > > > Thank you again for your thoughtful comments and suggestions. We hope we’ve addressed all your concerns. We are happy to clarify any additional questions you have and engage in a follow-up discussion.

---

### Official Review · Reviewer_jJu6 · 2025-11-08

**Soundness:** 3
**Presentation:** 1
**Contribution:** 2
**Rating:** 4
**Confidence:** 3

**Summary:**

The authors propose a method, CLEAR, for benchmarking energy consumption associated with individual components of LLMs during inference. One challenge noted by the authors is that the GPU power sensor refresh rate (dozens of milliseconds) is too coarse-grained to capture accurate readings of operations corresponding to components that take only microseconds to execute. In the CLEAR method, measurements are repeated sufficiently many times to reduce uncertainty. The authors observe a mismatch between FLOPs and energy consumption for different model components; for example, attention blocks consume more energy per FLOP than other types of modules.

**Strengths:**

1. Important problem, and simple method with relatively low barrier to adoption
2. A relatively extensive coverage of model architectures and families
3. Well-written related work section, even if coverage is not complete (see below)

**Weaknesses:**

1. line 218 ("Since no existing literature has yet provided the energy consumption of individual components..." -- the existence of IrEne (https://arxiv.org/abs/2106.01199) directly contradicts this. Not necessarily completely damning, but I would want to see explicit engagement with such a relevant related work, as well as an updated statement about the novelty and contribution claims the authors are making
2. 320-322: "We focus specifically on single-token generation to control for variability in output sequence length and to minimize cache based auto-regressive generation." -- This feels like a pretty significant limitation, as variability in output sequence length is emblematic of modern inference with language models, and the decode phase of generation has a different profile from the prefill phase. That being said, it would help if the authors could explain what precisely remains to be understood due to this limitation, or what findings are expected to generalize if any.
3. I am more than willing to update my rating given improvement/clarification, but it was extremely difficult to understand Figure 2 given that axes are unlabeled and even the main tables of measurements were confusing to understand without units directly in the tables
4. I see this as a writing quality issue more than a scientific soundness issue, but certain statements are written in a way that inadvertently creates unsubstantiated claims. One particularly glaring example is in

In-line comments:
1. line 038: earth.org source quotes a towardsdatascience blogpost that makes an extremely conservative estimate that doesn’t reflect the billions of daily queries that ChatGPT serves in reality. To be fair, the foresight of contextualizing the estimate alongside equivalencies is what quickly flagged the quoted cost as suspicious to me — please do just recheck the central figure quoted
2. line 163: I would want to see a citation for the 20-50ms figure (I think the authors include a citation elsewhere for similar information)
3. I would also prefer to see a citation and/or a specific ballpark range for the "significant amount of idle
energy drawn by CUDA" mentioned in line 174
4. organization of paragraph starting at line 364 could be improved for clarity — leading with the less obvious normalization layer observation can cause confusion if the more obvious overall trend (energy greater for float32) is not first established

**Questions:**

1. What findings are novel, vs what is a confirmation of previously established observations, vs common/general knowledge that does not have a formal source?
2. How exactly is the denominator obtained for the % Capture metric? i.e. how is $E_{model}$ measured? Please let me know if I missed this, but I was unable to find detailed methodology on this.

---

> ### Author Response · Authors · 2025-11-20
>
> We thank the reviewer for their constructive suggestions and for recognizing the importance of the problem we address. We appreciate the reviewer's acknowledgment of the soundness, simplicity, and the extensive coverage of different model families in our work.
>
> &nbsp;
>
> ### **W1. Comparative Discussion with Prior Work**
>
> A central contribution of CLEAR is a reliable methodology to precisely measure the true energy consumption of fine-grained Transformer components and its validation across 4 architecture families and 15 models. CLEAR consistently achieves >90% energy capture and exhibits high cross-trial consistency at both the layer-block level and the full-model level. Prior works to measure the energy consumption of computationally small components falls mainly in two categories:
>
> (a)  Methodologies that estimate energy at a coarse granularity often by measuring long sequences over large datasets and averaging to obtain per-token or fine-grained approximations [1][2][3]. For example, [3] reports the energy values over large datasets at the scale of MJ which fails to reliably isolate the true fine-grained effect at Attention level.
>
> (b) Hardware-centric approaches that rely on execution tracing tightly coupled with specific sensors or system instrumentation [4][5][6][7].  In such a methodology, access to physical GPU hardware becomes necessary and the additional sensor used needs to have low latency and high precision to reliably track/trace the function execution. Along with that, hardware based approaches are not scalable as they depend on specialized equipment and physical access to every target device making them difficult to deploy widely across large systems with diverse hardware environments.
>
> CLEAR departs from both paradigms by providing a component-level, software-operational method that is validated for completeness and consistency. As CLEAR operates entirely at the application layer / code module level (requiring only standard model execution and activation capture) it is straightforward to be adopted for other hardware devices and substantially broadens the experimental space for energy analysis.
>
> Previous works [1][8][9], do measure energy consumption using the two paradigms outlined above (a and b). However, these approaches explicitly acknowledge a key limitation: the coarse sampling frequency of available GPU/CPU energy sensors fundamentally restricts the granularity at which Energy Consumption can be attributed at the software or code-module level. As a result, they are unable to isolate the actual energy usage of microsecond-scale sub-operations or validate whether the reported estimates precisely reflect true energy consumption.
>
> CLEAR’s methodological rigor at an application/software level is not only novel in scope, but also uniquely practical and easily accessible for studying fine-grained energy behavior in modern Transformer architectures.
>
> We will incorporate the above discussion in the revised manuscript.
>
>
> [1] IrEne: Interpretable Energy Prediction for Transformers (https://arxiv.org/abs/2106.01199)
>
> [2] Green AI (https://arxiv.org/abs/1907.10597 )
>
> [3] Attentions Under the Microscope: A Comparative Study of Resource Utilization for Variants of Self-Attention (https://arxiv.org/abs/2507.07247 )
>
> [4] How Hungry is AI? Benchmarking Energy, Water, and Carbon Footprint of LLM Inference (https://arxiv.org/abs/2505.09598 )
>
> [5] PowerSensor3: A Fast and Accurate Open Source Power Measurement Tool (https://arxiv.org/abs/2504.17883 )
>
> [6] Understanding GPU Power: A Survey of Profiling, Modeling, and Simulation Methods (https://dl.acm.org/doi/10.1145/2962131 )
>
> [7] MLEE: Method Level Energy Estimation — A machine learning approach (https://www.sciencedirect.com/science/article/pii/S2210537921000822 )
>
> [8] CPPJoules: An Energy Measurement Tool for C++ (https://arxiv.org/abs/2412.13555 )
>
> [9] Accurate and Convenient Energy Measurements for GPUs: A Detailed Study of NVIDIA GPU’s Built-In Power Sensor (https://ieeexplore.ieee.org/document/10793163 )

---

> > ### Author Response · Authors · 2025-11-20
> >
> > ### **W2. Additional Experiments for Multi-Token Generation and Effect of KV Cache**
> >
> > We do acknowledge the reviewer’s concern about the limitation of single token generation and we conduct additional analysis to capture the Energy Consumption for Multi-token generation. Specifically, we conduct two analysis on Qwen2.5-3B model across varied Input Sequence Length:
> >
> > 1) We capture the energy consumption of the entire model across the Prefill phase, Decode phase, and complete Output Generation pipeline for a single new token. We observe that the Prefill stage and Output Generation [*using model.generate()*] show a linear increase in energy consumption and FLOPs with increasing input sequence length. However, the energy consumed by the Decode phase remains almost invariant to input sequence length due to KV cache reuse for additional token generation. KV cache eliminates redundant computation by retrieving previously computed Keys and Values in the Attention block. (Refer to Table 7 and Figure 5 in Appendix Section E.)
> >
> > 2) We explicitly measure the difference in energy consumption of the Attention block with and without the KV cache. When the KV cache is enabled, the Attention block exhibits nearly constant energy usage, since the model only processes the newly generated token while reusing previously stored Keys and Values. (Refer to Table 6 and Figure 5 in Appendix Section E.)
> >
> > We hope that the additional experiments provide a clearer resolution on the single token limitation identified in the previous draft. We have added the results and a more detailed discussion in Appendix Section E and will incorporate the discussion into the main text in future revisions of the manuscript.
> >
> > &nbsp;
> >
> > ### **W3. Presentational Details**
> >
> > We acknowledge that clearer axis labels and explicit units would greatly improve the readability of Figure 2 and the measurement tables. In the revised version of draft, we plan to add
> >
> > a) explicit axis labels for Figure 2, the X-axis will be labeled Trial Number (reflecting the temporal sequencing of measurements), and the Y-axis will be labeled as Energy Consumed (in mJ).
> >
> > b) measurement units in all table captions, specifying that energy values are in milliJoules (mJ),  all FLOP counts are reported in GFLOPs, $ E/FLOP $ and $ \Delta E / \Delta FLOP $ are expressed in mJ/GFLOP as mentioned in Section 4.2.
> >
> > &nbsp;
> >
> > ### **W4. Inline Comments**
> >
> > **InlineComment 1:  Carbon Footprint Estimate**
> >
> >
> > We rechecked the quoted figure and found that the source had limiting assumptions leading to a significant underestimate. We have updated the manuscript with stronger works that highlight the high environmental cost of LLMs, as reported by several key works:
> > [1] directly addresses the impact ofLLM inference. It estimates the 2025 annual carbon footprint of GPT-4o inference to be between 138,125 and 163,441 tons of CO2e based on an estimated per-Query energy (close to OpenAI’s disclosure) and an estimate of annual ChatGPT usage. The same paper projects a GPT-5 model on average energy consumption ranges from 2.33Wh for minimal reasoning to 17.15Wh for high reasoning.
> >
> > For comparison, the official report [2] states that a median Gemini query consumes 0.24 Wh equivalent to 0.03 $gCO_2e.$
> >
> >
> >
> > **InlineComment 2: Energy Sensor Sampling Rate**
> >
> > We have cross-validated the sampling rate of the NVML energy sensor using multiple independent sources [3][4][5][6]. We've already added the citations to the updated draft as well.
> >
> >
> > **InlineComment 3:  Measurement of Idle Energy**
> >
> > We experimentally measured the idle energy for Gemma3-4B (78.7 mJ/s) and Qwen2.5-3B (67.8 mJ/s) for 1 second time frame. Considering the standard sampling rate of nvml energy sensors (20-50ms as reported above), the idle energy is about 2-4 mJ and 1.5 - 3.5 mJ which accounts for about ~10% and approximately ~30% of the  Energy Consumption of Attention and Normalization block respectively.
> >
> > **InlineComment 4: Reorganising the paragraph**
> >
> > We acknowledge the reviewer's suggestion and have updated the manuscript accordingly.
> >
> > ---
> > [1]Measuring the environmental impact of delivering AI at Google Scale (https://arxiv.org/abs/2508.15734)
> >
> > [2]How Hungry is AI? Benchmarking Energy, Water, and Carbon Footprint of LLM Inference (https://arxiv.org/abs/2505.09598 )
> >
> > [3] Part-time Power Measurements: nvidia-smi’s Lack of Attention (https://arxiv.org/abs/2312.02741 )
> >
> > [4] Energy-Conscious LLM Decoding: Impact of Text Generation Strategies on GPU Energy Consumption (https://arxiv.org/html/2502.11723v2 )
> >
> > [5] Understanding GPU Power: A Survey of Profiling, Modeling, and Simulation Methods (https://dl.acm.org/doi/10.1145/2962131 )
> >
> > [6] High-Resolution Power Profiling of GPU Functions Using Low-Resolution Measurement (https://www.tu-chemnitz.de/informatik/PI/forschung/publikationen/download/LR_europar13.pdf)

---

> ### Author Response · Authors · 2025-11-20
>
> ###  **Q1. Detail on Implications and Contribution**
>
> a) The primary contribution of our work lies in the simple, easily adoptable methodology,  CLEAR for fine-grained energy measurement, together with its systematic validation and analysis. While prior works do measure the energy consumption of various models and different Attention mechanisms they fail to provide reliably measured component-level energy values nor a means to verify the correctness of measurements. Such lack of validation often results in unquantified errors and coarse approximations. In contrast, our empirical results demonstrate that CLEAR achieves >90% energy capture and exhibits high cross-trial consistency across diverse models.
>
>  b) While it is known informally that normalization layers internally cast FP16 activations to FP32 for numerical stability, prior work hasn't quantified the energy cost of the implicit typecast, nor examined its effect across models. Our analysis reveals that data-type conversion introduces a non-trivial and consistently measurable energy overhead, resulting in higher energy consumption for normalization layers in FP16 compared to FP32, despite the expectation that FP16 should always be more energy-efficient as displayed in other components like Attention, MLP and LM Head.
>
> c) To the best of our knowledge, empirical observations (RQ3 and RQ4) reveal effects that have not been documented in prior works on Transformer-based Energy measurement. Specifically
>
>  i) The energy-per-FLOP for the Attention mechanism is substantially higher than that of other components when measured in isolation.
> ii) The energy-per-FLOP decreases non-linearly with input sequence length, indicating the presence of fixed overheads and component-specific marginal energy coefficients.
>
> Above insights contradict the assumptions made in previous works:
>
>
> Prior works ([1][2]) assume that FLOPs directly represent computational effort and therefore correlate linearly with energy. However, FLOPs mainly encode parameterized computational load and not memory movements, synchronization, cache updates, or launch overheads. Any predictive methodology relying solely on FLOP counts such as Green AI’s [2] computational model using MUL and ADD costs, systematically underestimates attention-heavy workloads, especially for LLMs.
>
>
> IrEne ([3]) uses a regression based prediction model on feature dimensions to predict the energy consumption of the model. Since FLOPs are a deterministic function of the features dimensions (e.g., hidden size, head count, projection dims), such predictors cannot reliably capture fixed energy overheads ($ E_0 $) dependent on each model’s component implementation, code optimizations specific to hardware and multi-token energy estimation.
>
> d) Furthermore, we conducted additional experiments to isolate the energy impact of the KV cache, as well as the effects of various Attention mechanism variants and optimizations, which are presented in Appendix D and Appendix E, respectively.
>
> ### **Q2. Clarification for %Capture**
>
> As defined in Section 4.2.1, the denominator of the $ %Capture $ metric corresponds to the empirically measured total energy  consumed by the model [ $ E_{model} $ for %Capture (Model) ] or by a specific layer block [ $ E_{block} $ for %Capture (Block) ] during its complete execution. The total energy value in the denominator is obtained through a full forward pass, measured using the CLEAR methodology.
>
> Moreover, by grounding the denominator in the measured end-to-end energy of the model or block (rather than a Theoretical or Function Tracing based estimate), the $ %Capture $ metric provides a faithful indicator of how comprehensively CLEAR accounts for the true energy footprint. The formulation enables reliable comparison across components and architectures, since both the numerator and denominator are measured under identical execution and instrumentation conditions.
>
> ---
>
> [1] Estimating the Energy Footprint of Software Systems: a Primer (https://arxiv.org/abs/2407.11611)
>
> [2] Green AI (https://arxiv.org/abs/1907.10597)
>
> [3] IrEne: Interpretable Energy Prediction for Transformers (https://arxiv.org/abs/2106.01199)
>
> ---
>
> Thank you again for your thoughtful comments and suggestions. We hope we’ve addressed all your concerns. We are happy to clarify any additional questions you have and engage in a follow-up discussion.

---

### Author Response · Authors · 2025-12-03
**Overall Summary**

We thank the reviewers for their thoughtful and constructive evaluations. To address the major concerns raised, we conducted **three sets of new experiments:**

&nbsp;

 - **Multi-token generation experiments** measuring the energy of the **prefill, decode**, and **full generation** stages across different input sequence lengths. These clarify how CLEAR’s single-token results generalize and show that decode stage energy consumption remains nearly constant due to KV-cache reuse. (Refer Appendix D of updated manuscript)
 - **KV-cache isolation analysis** comparing energy consumed by Attention Mechanism **with and without KV cache**. The experimentation specifically validates whether CLEAR can reliably capture memory movements and its adoption to multi-token generation setting (Refer Appendix E for detailed analysis)
 - **Attention-variant experiments** evaluating **Flash Attention, SDPA**, and **Eager Attention**, as well as runtime optimizations such as Torch Compile, Max Autotune and Reduce Overhead. These experiments strengthen the practical relevance of CLEAR by showing how different implementations and optimizations change energy consumption.

&nbsp;

We also improved presentation quality with clearer figures, explicit citations, revised related work, and detailed methodological explanations.
Across the reviews, there is strong agreement that the paper addresses a timely and important problem, that the methodology is sound and adoptable, and that the component-level insights are valuable for the community. Three reviewers (R1, R2, R4) place the work near the acceptance threshold, and R3 already recommends acceptance. During the rebuttal period, we missed out on gaining valuable feedback from the reviewers R1,R2 and R3 on our official comments; however, we are confident that we had addressed major concerns regarding the experimentation as well as the implicational value of our approach and given time, would have convinced the reviewers for updating the scores.

---

> ### Author Response · Authors · 2025-12-03
>
> Below we outline how each reviewer’s concerns were addressed:
>
>
> ## **Reviewer jJu6 (R1)**
> R1 requested clearer engagement with related work (especially IrEne), clarification of the single-token limitation, clearer figures/tables, and explanation of the %Capture metric.
>  We addressed all points by adding multi-token and KV-cache experiments, correcting carbon references, expanding citations on sensor sampling and idle energy, and improving figure/table clarity with explicit labels and units. **R1 noted willingness to adjust their score once clarifications are provided; all items have now been incorporated.**
>
> &nbsp;
>
> ## **Reviewer AfSZ (R2)**
> R2 requested clarification on **novelty, robustness, and practical usefulness**. All three concerns are fully resolved in the revised manuscript. We use Flash Attention for our experiments and later, we compare it with other attention mechanisms separately.
>
> &nbsp;
>
> ### **Novelty**
>  We now clearly distinguish the methodological contribution (a validated component-level pipeline with >90% capture across 15 models) from the new empirical findings uncovered by CLEAR (attention’s higher energy-per-FLOP, non-linear scaling with sequence length, FP16 normalization overhead). This directly answers R2’s concern for a clearer contribution statement.
>
> &nbsp;
>
> ### **Robustness**
>  R2 asked whether CLEAR behaves reliably in realistic inference settings primarily due to KV Cache. We added experiments showing:
>  - prefill energy does **not** scale linearly with sequence length or FLOPs,
>  - decode energy stays **nearly constant** due to KV-cache reuse,
>  - a small increase in long-sequence energy arises from **growing KV tensors**, and
>  - Energy consumed by Attention Mechanism behaves as expected **with vs. without KV** cache.
> These experiments confirm CLEAR’s reliability under real execution conditions.
>
> &nbsp;
>
> ### **Practical usefulness**
> R2 asked how CLEAR can guide actual model architecture design decisions. We added:
>  - comparisons of **Flash Attention, SDPA, and Eager Attention**, showing SDPA ≈ FlashAttn < Eager,
>  - quantified effects of widely used optimizations (Torch Compile, Max Autotune, Reduce Overhead), which reduce energy by **30–64%**,
>  - and an explanation of how CLEAR provides the **ground-truth measurements** needed for predictive models and future energy-aware architecture design needed by existing related works.
> These results directly demonstrate the practical value R2 asked for.
>
>
> **We emphasize that all of R2’s questions, especially those related to practical applicability have been fully addressed.**
>
> &nbsp;
>
> ## **Reviewer uuyn (R3)**
> R3 is strongly supportive along with familiarity to the related works (Confidence of **5** in our work), noting the methodological soundness, breadth of evaluation, and clarity of presentation.
>
> &nbsp;
>
> ## **Reviewer vHyf (R4)**
> R4 questioned methodological originality. We respectfully note that very few prior works offer any practical, component-level methodology for Transformer energy measurement, leaving the community without a foundation for principled energy-aware design for transformer architectures. CLEAR fills that foundational gap by providing component isolation, activation replay, energy amplification, and systematic validation, capabilities previously unavailable without specialized hardware.
> In addition, responding directly to R4’s request for actionable tooling, we have prepared a **Python package** for CLEAR [1], allowing industry teams and practitioners to directly apply the methodology. **The Python Package allows CLEAR to be an immediately usable contribution.**
>
> &nbsp;
>
> [1] **CLEAR Package**: https://anonymous.4open.science/r/CLEAR-D487/package/README.md
>
> &nbsp;
>
>
> With all major concerns having been fully addressed, the revised paper now offers:
>  (i) a validated, practical, model-agnostic 3 step pipeline for fine-grained energy measurement,
>  (ii) new and previously undocumented empirical findings, and
>  (iii) an actionable software tool for the community.

---

### Meta-Review · Area_Chair_7ctv · 2025-12-25

**Summary:**

* Reviewers agree the paper addresses an important and timely problem and presents a component-level energy measurement pipeline.
* The empirical evaluation is thorough and several observations are useful, but the core contribution is perceived as an engineering methodology rather than a novel scientific contribution. While the rebuttal and added appendices address multiple important concerns, they do not fundamentally change the contribution strength.
* Key concerns, particularly around overstated novelty claims and insufficient engagement with close related work, remain unaddressed in the PDF. It is plausible some of these concerns would have been resolved through further discussion under normal review conditions.
* Based on the partial resolution of issues in the rebuttal and the remaining open points, reviewers would likely not have reached a clear consensus in favor of acceptance.

**Reviewer Concerns:**

Reviewer jJu6
* [Addressed] Multi-token + KV cache in Appendix E, Figure 2 units, sampling rate citations, clearer %Capture denominator definition, updated intro carbon-footprint framing.
* [Not addressed] Citation/discussion of IrEne and other close related work in the updated PDF, and the rebuttal’s explicit idle energy / CUDA quantified measurements. The claims that bothered the reviewer most (overstating contributions, related work) are not addressed in the updated PDF, only discussed on OpenReview.

Reviewer AfSZ
* [Addressed] The “repeated measurements are obvious” critique is mitigated by clearly framing CLEAR as a pipeline, KV-cache / production-style effects is addressed in Appendix E, the implications for efficient design request is addressed in Appendix D.
* [Not addressed] The paper still does not provide a concrete “better-than-FLOPs” estimator that avoids measurement, the "does cost change across different hardware?" question is not really addressed.

Reviewer uuyn
* [Addressed] Implications are partially addressed in Appendix D and in the rebuttal.
* [Not addressed] Missing related work, test multiple sizes of the same model family.

Reviewer vHyf
* [Addressed] CLEAR is “just averaging” is partially addressed by reframing CLEAR as a pipeline, question on implementation details addressed in Appendix D.
* [Not addressed] The "lack of scientific contribution" remains only rhetorically answered in the rebuttal: the manuscript still does not introduce a new theory, metric, or design principle, nor does it derive concrete architecture-level prescriptions or hyperparameter guidance, hardware generalization concern is not empirically addressed.

**Reviewer Scores:**

Reviewer jJu6 - the rebuttal would likely have led to further discussions, and given that the authors would have revised their related work and claims, may have led to a score increase --> 4 or 6.

Reviewer AfSZ - I believe the reviewer would have kept their score --> 4.

Reviewer uuyn - The reviewer wouldn’t have adjusted the score --> 8.

Reviewer vHyf - I believe the reviewer would have kept their score --> 4. The reviewer writes "method too simple to justify a conference paper".

---

### Decision · Program_Chairs · 2026-01-26

Reject